# Unsupervised machine learning reveals risk stratifying glioblastoma tumor cells

Nalin Leelatian[1,2,3†], Justine Sinnaeve[1,2†], Akshitkumar M Mistry[2,4], Sierra M Barone[1], Asa A Brockman[1,2], Kirsten E Diggins[1,2], Allison R Greenplate[2,3], Kyle D Weaver[4], Reid C Thompson[4], Lola B Chambless[4], Bret C Mobley[3], Rebecca A Ihrie[1,2,4*], Jonathan M Irish[1,2,3*]

[1]Department of Cell and Developmental Biology, Vanderbilt University, Nashville, United States; [2]Vanderbilt-Ingram Cancer Center, Vanderbilt University Medical Center, Nashville, United States; [3]Department of Pathology, Microbiology and Immunology, Vanderbilt University Medical Center, Nashville, United States; [4]Department of Neurological Surgery, Vanderbilt University Medical Center, Nashville, United States

**Abstract** A goal of cancer research is to reveal cell subsets linked to continuous clinical outcomes to generate new therapeutic and biomarker hypotheses. We introduce a machine learning algorithm, Risk Assessment Population IDentification (RAPID), that is unsupervised and automated, identifies phenotypically distinct cell populations, and determines whether these populations stratify patient survival. With a pilot mass cytometry dataset of 2 million cells from 28 glioblastomas, RAPID identified tumor cells whose abundance independently and continuously stratified patient survival. Statistical validation within the workflow included repeated runs of stochastic steps and cell subsampling. Biological validation used an orthogonal platform, immunohistochemistry, and a larger cohort of 73 glioblastoma patients to confirm the findings from the pilot cohort. RAPID was also validated to find known risk stratifying cells and features using published data from blood cancer. Thus, RAPID provides an automated, unsupervised approach for finding statistically and biologically significant cells using cytometry data from patient samples.

*For correspondence:
rebecca.ihrie@vanderbilt.edu
(RAI);
jonathan.irish@vanderbilt.edu
(JMI)

†These authors contributed equally to this work

## Introduction

A modern goal of quantitative analysis of single cell data in human cancers is to move beyond human-driven identification of cell types using known markers (expert gating) to machine learning tools that can reveal and characterize novel and abnormal cells (*Diggins et al., 2015*; *Greenplate et al., 2019*; *Irish, 2014*; *Saeys et al., 2016*). Citrus, an automated analysis tool based on assignment of samples to binary categories (e.g. 'healthy' and 'disease') before testing whether cell populations are associated with these categories, was designed with this purpose in mind (*Supplementary file 1*; *Bruggner et al., 2014*). However, many important clinical features of patient tissue samples are reported as continuous variables, such as time to progression, overall survival, or percentage of immune infiltrate, which can be challenging to convert to arbitrary binary categories and may not be driven by a single unified cellular phenotype (*Gonzalez et al., 2018*; *Good et al., 2018*; *Levine et al., 2015*). Similarly, known, healthy cell populations from different stages of development or differentiation may be required for some approaches, such as developmentally dependent predictor of relapse (DDPR *Good et al., 2018*), and are not always available or fully represented for all datasets. This is especially acute for some tissues, such as brain, which may be quiescent in adults and not routinely sampled in clinical care or research. Tools are needed that can take into account continuous clinical variables that may be censored, such as overall survival or progression free survival (PFS), and which operate in an unsupervised manner. Ultimately, tools that

work with high dimensional data should help users to translate findings from an algorithmic machine learning tool to common practice by identifying lower dimensional correlates that can be used to validate signatures using a complementary, clinically tractable approach. This transparency was a focus of the tool design and validation strategy used here. A computational workflow constructed for this purpose should also be validated via repeated subsampling of data to ensure the phenotypes identified are robust, by testing of different dimensionality reduction tools, by testing across multiple datasets, and by validation of prognostic signatures using complementary approaches. Finally, a practical challenge of modern single cell discovery projects is that they may often be at a project point where they are working with a smaller initial cohort (around 25 patients). This study size is powered to closely correlate cell subsets with patient outcomes using signaling cytometry data, as this study and others have shown for blood cancers (*Gonzalez et al., 2018*; *Good et al., 2018*; *Irish et al., 2004*; *Irish et al., 2010*; *Kotecha et al., 2008*; *Levine et al., 2015*), but necessitates extensive statistical and biological validation, as discussed below.

RAPID (Risk Assessment Population IDentification) is a newly created algorithm that was designed using single cell cytometry data and which addresses the key challenges of clinical research using discovery cohorts of patients (https://github.com/cytolab/RAPID; *Leelatian, 2020*; copy archived at https://github.com/elifesciences-publications/RAPID). This open-access tool can couple single cell experiments to clinical outcome and other variables in an unsupervised manner and provide information that can be translated into simplified tests on other platforms. For this study, the algorithm was assessed for 1) cluster stability (*Melchiotti et al., 2017*) for both cells and phenotypes; 2) modularity (*Diggins et al., 2015*; *Saeys et al., 2016*), which would allow the algorithm to function with a range of dimensionality reduction approaches, such as no dimensionality reduction, t-distributed stochastic neighbor embedding (t-SNE *Amir et al., 2013*), or uniform manifold approximation and projection (UMAP *Becht et al., 2019*), clustering tools, such as FlowSOM (*Van Gassen et al., 2015*) or dbscan (*Akers et al., 2013*), and enrichment analysis tools, such as marker enrichment modeling (MEM *Diggins et al., 2017*); 3) transparency, evaluated as the ability to derive simple models of data structure (*Gandelman et al., 2019*), such as decision trees or flow cytometry gating hierarchies, so that new datasets could be easily assessed; 4) independence - whether risk stratifying cell populations are independent of known predictors (age, others); and 5) reproducibility and translational potential, tested by gathering additional data using traditional, one-dimensional immunohistochemistry (IHC) that is widely used in clinical testing.

Here, the utility and validity of the RAPID algorithm were tested using two datasets with varying levels of prior knowledge, numbers of patients and cells, and outcome trajectories. The first was a new data set of 28 glioblastoma patient samples and is described in detail below. Central findings from this first dataset were then validated using 73 additional samples analyzed using a different technology. The second was a previously published data set of 54 bone marrow samples from B cell precursor acute lymphoblastic leukemia (*Good et al., 2018*). This study was chosen as an example of a dataset in which prognostic features had already been independently identified, and so validation was assessed by whether known features were revealed by RAPID.

When applied to single cell cytometry data from human tumors, as shown here, the aim of RAPID was to reveal and characterize populations of risk stratifying cells. For this goal, glioblastoma, the cancer type in the first dataset, represents an excellent challenge, since glioblastoma is a highly heterogeneous solid tumor that is amenable to single cell approaches (*Doxie et al., 2018*; *Gonzalez et al., 2018*; *Greenplate et al., 2019*; *Leelatian et al., 2017a*) and where there is a great opportunity for molecular prognostic features to have an impact on new treatments and clinical care. Glioblastoma is the most common primary brain tumor in adults, is highly aggressive, and is known to contain cells with diverse genomic and transcriptomic features reflecting abnormal neural lineages (*Bhaduri et al., 2020*; *Leelatian et al., 2017a*; *Ostrom et al., 2017*; *Patel et al., 2014*; *Wei et al., 2016*). Previous studies in glioblastomas have either measured signaling states in bulk primary tumors (*Brennan et al., 2009*; *Brennan et al., 2013*; *Verhaak et al., 2010*) or characterized genomic and transcriptomic profiles in a limited number of single cells (<33,000) (*Bhaduri et al., 2020*; *Johnson and White, 2014*; *Neftel et al., 2019*; *Patel et al., 2014*; *Stommel et al., 2007*; *Wei et al., 2016*). While differing subclasses of glioblastomas were proposed a decade ago (*Verhaak et al., 2010*), these categories do not correspond to large differences in prognosis and are not always reflected by individual cells (*Patel et al., 2014*). Mosaic amplifications of receptor tyrosine kinase (RTK) genes are commonly observed in subsets of cells within a single glioblastoma

tumor (*Snuderl et al., 2011*), suggesting that single cell analysis of glioblastoma should include signaling measurements. In other cancer types, phospho-protein signaling has repeatedly revealed cancer cell subsets that are closely linked to patient clinical outcomes (*Gonzalez et al., 2018*; *Good et al., 2018*; *Irish et al., 2004*; *Irish et al., 2010*; *Kotecha et al., 2008*; *Levine et al., 2015*). These results suggest that a protein-level approach in a small pilot cohort may reveal phenotypically distinct cancer cell subsets whose abundance provides new ways to stratify glioblastoma outcomes. While it is known that upstream regulators of pro-growth and pro-survival signaling are altered in brain tumors, little is known about the activation states of signaling effector proteins in single glioblastoma cells, as these features are inaccessible to sequencing modalities (*Meyer et al., 2015*; *Mistry et al., 2019*; *Snuderl et al., 2011*; *Spitzer and Nolan, 2016*).

Another challenge that the RAPID algorithm was designed to address was the need to work with heterogeneous cell phenotypes and populations that might be rare and variable across patients. Cytometry data are a good match for this type of algorithm, as a large number of cells are collected from each tumor sample, the data have an excellent signal-to-noise ratio and support quantitative comparisons, and cytometry enables direct measurement of signaling pathway activation (*Irish et al., 2010*; *Kotecha et al., 2008*; *Mistry et al., 2019*; *Myklebust et al., 2017*). When glioblastoma mass cytometry data were analyzed by RAPID, both negative- and positive-prognostic phenotypes were identified, with protein-level phenotypes not described by prior studies. Statistical description of prognostic phenotypes within the RAPID algorithm then enabled the design of a simple workflow using traditional IHC, which stratified outcome in a separate set of 73 glioblastoma patient tissues.

## Results

### RAPID identifies stratifying cell subsets in an automatic and unsupervised manner

The RAPID algorithm workflow is depicted in *Figure 1* using results from Dataset 1. Following patient-specific identification of major cell types (*Figure 1a*), the algorithm (*Figure 1b*) randomly sampled an equal number of glioblastoma cells from each patient's tumor and analyzed the cells on a single, common t-SNE. This even sampling was conducted to generate a t-SNE analysis where each patient contributed equally. Subsequent statistical testing (*Figure 1c*) included repeated subsampling to ensure that sampled cells were representative of the original tumors. After multiple statistical tests, the most robust and reproducible cell types identified by RAPID were validated biologically, including using a new data type and a larger cohort (*Figure 1d*).

The RAPID algorithm was unsupervised and included two key statistical decisions. The first decision was the automation of the number of target clusters sought at the clustering step (*Figure 1b*, middle). This was achieved through repeated analysis with the chosen clustering tool, in this case FlowSOM (*Van Gassen et al., 2015*), followed by statistical analysis. RAPID iteratively tested a range (cluster number 5–50) of unsupervised self-organizing maps from FlowSOM to identify an appropriate number of stable clusters containing phenotypically homogenous cells. The minimum number of clusters that minimized intra-cluster variance for each feature was calculated after all iterations were completed and set as the optimized target cluster number (see Materials and methods). Clustering with other tools, such as DBSCAN, or clustering on untransformed axes, was both slower and less accurate in identifying stable, phenotypically distinct clusters, consistent with published observations (data not shown and *Weber and Robinson, 2016*). The second decision was in assessing cluster abundance in patients (*Figure 1b*, right). RAPID assigned patients to high or low abundance for each automatically identified cluster based on a statistical cut point, set as the interquartile range of the population abundance across the samples (see Materials and methods). These two decisions resulted in automation of steps that are typically manual in cytometry analysis.

After finding clusters in an unsupervised manner and determining which patients' tumors contained a high level of each cluster, the last step in a run of RAPID was to test whether each cluster stratified risk of death. For this last test, RAPID applied a univariate Cox survival analysis to determine the correlation between the abundance of tumor cells in each cluster and patient survival outcome (*Supplementary file 2*). Clusters were identified as prognostic by assessing the hazard ratio (HR) of death in patients who had either high or low abundance of the cell cluster. Negative and

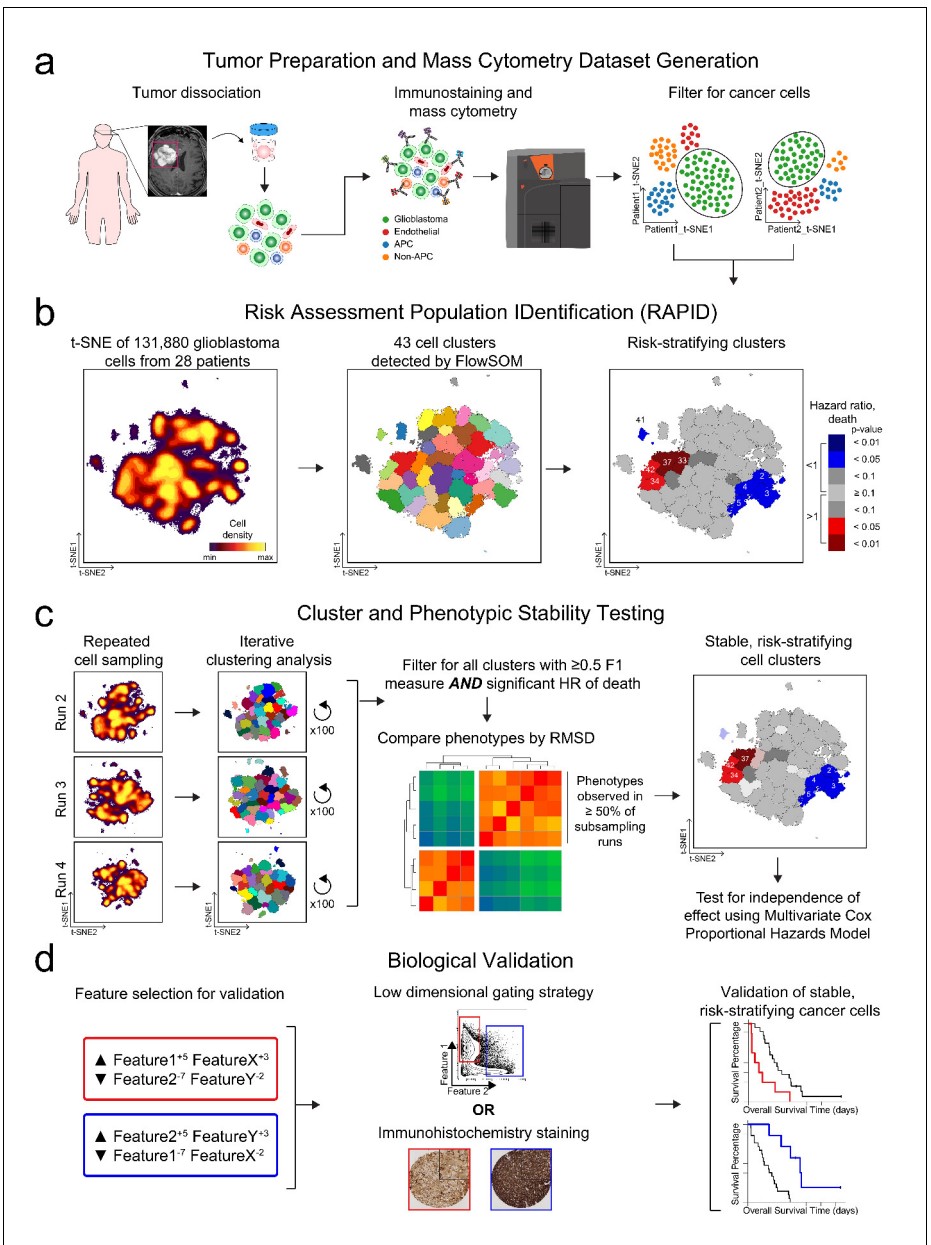

**Figure 1.** RAPID identifies single cell phenotypes associated with continuous clinical variables that are stable and validated via complementary approaches. (a) Graphic of tumor processing and data collection. After data collection and standard pre-processing, non-immune, non-endothelial glioblastoma cells were computationally isolated for analysis by RAPID. (b) RAPID workflow on glioblastoma cells identified from 28 patients and computationally pooled for t-SNE analysis. Cell subsets were automatically identified by FlowSOM and were systematically assessed for association with patient overall or progression-free survival. 43 glioblastoma cell subsets were identified and were color-coded based on hazard ratio of death and p-values (HR >1, red; HR <1, blue). Cell density, FlowSOM clusters, and cluster significance are depicted on t-SNE plots. (c) RAPID results were tested for stability. Each tumor was randomly subsampled for 4,710 cells multiple times. Each of these cell subsampling runs was subject to 100 iterative FlowSOM analyses and an F-measure was calculated for each cluster. Only clusters with an F-measure of greater than 0.5 were considered stable. Then, the phenotypes of stable clusters associated with patient outcome were assessed via RMSD and used to determine stable phenotypes. (d) Validation of the findings from the mass cytometry data was done using lower dimensional gating strategies and an orthogonal technology to confirm the biological findings.

The online version of this article includes the following figure supplement(s) for figure 1:

*Figure 1 continued on next page*

*Figure 1 continued*

**Figure supplement 1.** Single cell quantification of identity proteins and phospho-protein signaling in glioblastoma.

**Figure supplement 2.** Quantitative MEM labels of the enriched identity proteins and signaling features of all glioblastoma cell subsets identified by RAPID.

**Figure supplement 3.** Glioblastoma cell subsets showed differential enrichment of identity proteins and phosphorylated signaling effectors.

**Figure supplement 4.** Divergent phenotypes are associated with patient outcomes.

**Figure supplement 5.** Abundance of immune cells correlated with the abundance of prognostic cell subsets.

**Figure supplement 6.** RAPID identified four populations associated with time to disease progression.

positive prognostic clusters were colored red or blue, respectively, if they were significantly associated (p<0.05) with an HR that was >1 (negative, red) or <1 (positive, blue). The RAPID algorithm used statistical analysis of the common t-SNE, feature variance, and population abundance to automatically set all computational analysis parameters, independent of clinical outcomes.

The output of RAPID includes a PDF containing a color-coded, 2D t-SNE plot depicting all Flow-SOM clusters, a 2D t-SNE plot colored by clusters which were significantly associated with patient outcome, and Kaplan-Meier survival plots of patients for each subset (additional files described in Materials and methods) (*Figure 1b*). To compactly report and depict the phenotype of algorithmically identified cell subsets, RAPID used Marker Enrichment Modeling (MEM) labels (*Diggins et al., 2017*). Thus, feature enrichment was reported on a +10 to −10 scale, where +10 indicated that the feature was especially enriched in those cells and −10 indicated that the feature was specifically excluded from those cells, relative to all other cells in other clusters. The MEM label here was thus an objective description of what made each population distinct from the other clusters identified by RAPID. In summary, RAPID provided an unsupervised, automated, statistical approach to revealing and characterizing clinically significant cells.

## Identification of risk stratifying glioblastoma cells in Dataset 1

RAPID was designed for datasets like Dataset 1, a pilot glioblastoma mass cytometry dataset including cells collected from 28 patients with *isocitrate dehydrogenase (IDH)* wild-type glioblastoma at the time of primary surgical resection (*Supplementary file 3*). This dataset is currently available online (https://flowrepository.org/id/FR-FCM-Z24K). The median PFS and overall survival (OS) after diagnosis were 6.3 and 13 months, respectively, typical of the trajectory of this disease (*Stupp et al., 2005*). Resected tissues were immediately dissociated into single cell suspensions as previously reported (*Leelatian et al., 2017b*) and the resulting cells were stained with a customized antibody panel, which was designed to capture the expression of known cell surface proteins, intracellular proteins, and phospho-signaling events (*Supplementary file 4*). Collectively, the antigens included in this panel positively identified >99% of viable single cells within any given tumor sample (see Materials and methods). To identify glioblastoma cells prior to RAPID, as in *Figure 1a*, a patient-specific t-SNE was created using 26 of the measured markers for the tumor and stromal cells from each patient's tumor (*Amir et al., 2013*; *Figure 1—figure supplement 1* and *Supplementary file 4*). Patient-specific t-SNE maps revealed non-glioblastoma populations of immune (CD45$^+$) and endothelial (CD45$^-$CD31$^+$) cells, consistent with prior mass cytometry and sequencing studies of gliomas (*Diggins et al., 2017*; *Greenplate et al., 2019*; *Leelatian et al., 2017a*; *Neftel et al., 2019*; *Patel et al., 2014*). Immune and endothelial cells from each individual patient were computationally excluded prior to subsequent downstream analysis (*Figure 1*, *Figure 1—figure supplement 1*), and CD45$^-$CD31$^-$ cells were labeled as glioblastoma cells.

Plots of cell density on the t-SNE axes revealed phenotypically distinct subpopulations of glioblastoma cells within a single patient's tumor (example patient LC26: *Figure 1—figure supplement 1*, maps for all patients: *Supplementary file 6*) Intra-tumoral subsets were distinguished by differences in expression of core neural identity proteins and by aberrant co-expression of neural lineage and stem cell proteins. In the example case of tumor LC26, abnormal phenotypes in glioblastoma cells included co-expression of astrocytic S100B and stem-like CD133 or co-expression of markers associated with different molecular subtypes of glioblastoma, such as mesenchymal (CD44) and classical (EGFR) (*Figure 1—figure supplement 1*; *Verhaak et al., 2010*). These results with protein

confirmed the existence of non-canonical cell types that had previously been observed in single-cell RNA-seq (*Patel et al., 2014*). The abnormal co-expression of identity proteins seen here, as well as previously reported single cell studies relying on inferred DNA alterations (*Neftel et al., 2019*), indicate that the large majority of the CD45⁻CD31⁻ cells were likely cancer lineage cells.

Using an equal number of subsampled glioblastoma cells from each patient (see Materials and methods), a single, common t-SNE map was created to represent glioblastoma cell protein phenotypes across all patients (N = 131,880 cells; 4,710 cells x 28 patients, using 24 measured features). The RAPID algorithm, using the pooled data from all patients, identified 43 phenotypically distinct cell clusters, and then determined for each tumor whether a patient was high or low for a particular cluster using the interquartile range of abundance for that cluster. For example, for glioblastoma cluster 24, the interquartile range was 0.67% to 3.36%, resulting in a cut point of 2.69%. Those patients with ≤2.69% were designated 'low' for cluster 24 while those with >2.69% were assigned to the 'high' group. Additional cut points, based on splitting populations into quartiles or tertiles, were tested and resulted in consistent prognostic phenotypes (the average F-measure of patients being consistently assigned to the high, low, or neither categories identified below was 0.86). The number of tumors that contributed to each cluster varied between the 43 clusters, but a median of 8 tumors contained cells in each cluster (*Supplementary file 2*, *Supplementary file 6*). Furthermore, each cluster contained cells from at least 4 tumors and, at the median, contained cells from 12 tumors (*Supplementary file 5*, *Supplementary file 6*).

The RAPID algorithm identified four Glioblastoma Negative Prognostic (GNP) clusters (red; clusters 33, 34, 37, and 42) and five Glioblastoma Positive Prognostic (GPP) clusters (blue; clusters 2, 3, 4, 5, and 41) whose abundance was associated with overall survival (*Figure 1b*). MEM labels were used to identify the enriched features of risk stratifying glioblastoma cells (*Figure 1—figure supplements 2* and *3*). MEM labels were calculated for both total proteins (P), such as S100B and EGFR, and signaling effectors (S), such as p-STAT5. GNP cells aberrantly co-expressed neural-lineage proteins (astrocytic S100B and stem-like SOX2). Additionally, GNP cells displayed phosphorylation of RTK signaling effectors known to promote cell survival, growth, and proliferation (e.g. p-STAT5$^{Y694}$, p-S6$^{S235/S236}$, p-STAT3$^{Y705}$) (*Figure 1—figure supplements 2* and *4*). The MEM protein enrichment values (average and standard deviation) for GNP cells included neural lineage determinants (▲S100B$^{+5\pm1.6}$, SOX2$^{+5\pm1}$) and phospho-proteins (▲p-STAT3$^{+3\pm2.1}$, p-STAT5$^{+2\pm1.8}$, p-S6$^{+3\pm1.4}$) and identified proteins that were specifically lacking in GNP cells relative to other glioblastoma cell clusters (▼EGFR$^{-2\pm0.1}$, GFAP$^{-4\pm0.7}$, CD44$^{-4\pm0}$) (*Figure 1—figure supplement 4*). In contrast, GPP cells were positively enriched for EGFR (▲EGFR$^{+5\pm0.8}$) and consistently lacked pro-survival phospho-proteins (▼p-S6$^{-4\pm3.7}$, p-STAT5$^{-2\pm0.8}$, p-STAT3$^{-2\pm1.6}$) and one of the proliferation markers measured (▼cyclin B1$^{-3\pm3.3}$) (*Figure 1—figure supplement 4*).

Non-malignant cells, including immune and endothelial cells, were excluded from initial RAPID analyses and subsequent biaxial gating confirmed that the GNP and GPP subsets were not unexpected residual CD45⁺ or CD31⁺ cells (*Figure 1—figure supplement 4*). However, infiltrating immune cells can comprise a large proportion of non-cancer cells in glioblastomas and have highly variable overall abundance across patients (*Hussain et al., 2006*). Notably, GPP-high (n = 7) patients' tumors all contained more than 9% CD45⁺ cells (median %=25.3 ± 13.8), whereas all GNP-high (n = 8) patients' tumors contained less than 9% CD45⁺ cells (median %=3.3 ± 2.4, p<0.001, *Figure 1—figure supplement 5*, *Supplementary file 2*).

## Identification of risk stratifying B-cell leukemia cells in Dataset 2

FCS files from a previously published mass cytometry study of B-cell precursor acute lymphoblastic leukemia (BCP-ALL) by an independent lab were input into the RAPID workflow to test whether the RAPID algorithm could re-discover prognostic cell subsets in other disease settings (*Good et al., 2018*). Dataset 2 is available online (originally: https://github.com/kara-davis-lab/DDPR/releases, in this study: https://github.com/cytolab/RAPID). This dataset contained almost twice the number of patients (n = 54) but less than half the number of total cells compared to Dataset 1 (48,600) because of a single patient with only 900 live, lineage-negative blast cells (*Good et al., 2018*). A total of 47 clusters were identified by RAPID, 3 of which were negative prognostic cell subsets that were associated with time to relapse (*Figure 2*). Importantly, features identified in the original publication as part of the signature associated with relapse (black text, *Figure 2*) were re-identified using RAPID. In the protein feature MEM values, enrichment of CD38 and CD34 was consistent with previously

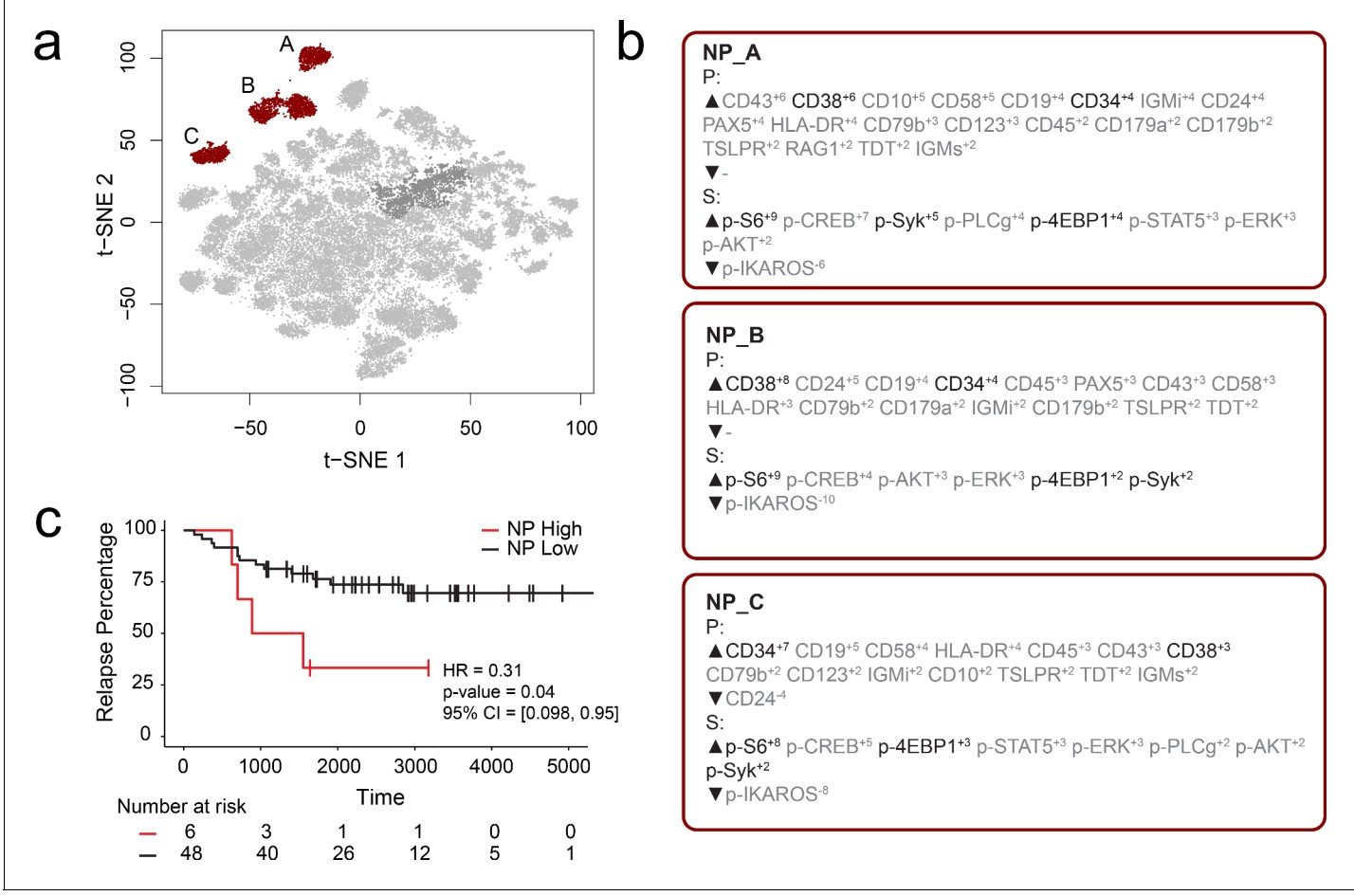

**Figure 2.** RAPID analysis of a published B-cell leukemia dataset to identify negative prognostic cell subsets. (**a**) t-SNE plot of 54 B-cell leukemia patient samples with negative prognostic populations (A, B, C) colored in red. (**b**) MEM labels for three negative prognostic cell subsets (NP_A, NP_B, NP_C). Features important in the original discovery of predictors of relapse are colored in black. (**c**) Kaplan-Meier Curve comparing time to relapse in patients with high abundance of negative prognostic cells (identified by RAPID) to patients with low abundance of negative prognostic cells.

reported trends in pre-pro B cell-like phenotypes in BCP-ALL. Most notably, the signaling features p-S6, p-SYK, and p-4EBP1, which were important features positively associated with relapse in the DDPR model, were enriched in the negative prognostic populations identified by RAPID. Thus, RAPID was able to identify cells and features associated with time to relapse in another disease setting, generating a signature of negative-prognostic cells consistent with the original findings by another research group.

## Statistical validation 1: Clusters identified by RAPID were statistically robust

To determine the stability of the clusters identified by RAPID, 99 additional runs of FlowSOM were performed within the RAPID workflow (*Figure 1c*). Due to the stochastic nature of FlowSOM, the clusters identified in each subsequent run could contain different cells. For each of the clusters, an F-measure was calculated, based on the accuracy of cell assignment within a cluster in subsequent iterations of FlowSOM (see Methods, *Supplementary file 2*). Of the original 43 clusters, five had an average F-measure of less than 0.5 (average F-measure of all clusters = 0.75). These five clusters, including cluster 33, previously identified as a GNP cluster, were considered unstable and were not included in subsequent analyses (indicated by shading in *Figure 1* and *Figure 1—figure supplement 2*, and asterisks in *Figure 1—figure supplement 3* and *Supplementary file 2*).

## Statistical validation 2: Clusters identified by RAPID were not dependent on individual patients or sub-samplings

A key design decision in RAPID was the use of an equal number of cell events from each patient to avoid tumors disproportionately impacting the analysis based on the number of cells collected. However, this decision limits a given RAPID analysis run to a number of cells equal to the smallest collected from any one patient. For the tumors studied here, the number of live glioblastoma cells ranged from 4,710 to 330,000 cells per patient. To test whether the cells subsampled for RAPID were representative of the total tumor sample and eliminate the possibility that randomly subsampled cells from larger samples are not representative, 9 additional t-SNE analyses were generated, each with a different sample of 4,710 cells selected at random, with replacement, from each patient. Each of these 9 t-SNE projections was then used in a new RAPID analysis, creating 10 total analyses (the original and 9 new tests). Of these, a total of 55 clusters from the 10 runs were considered stable (F-measure >0.5) and prognostic (see Methods, *Figure 3*). An F-measure could not be calculated on a cell-by-cell basis because the cells varied between analyses, but the average

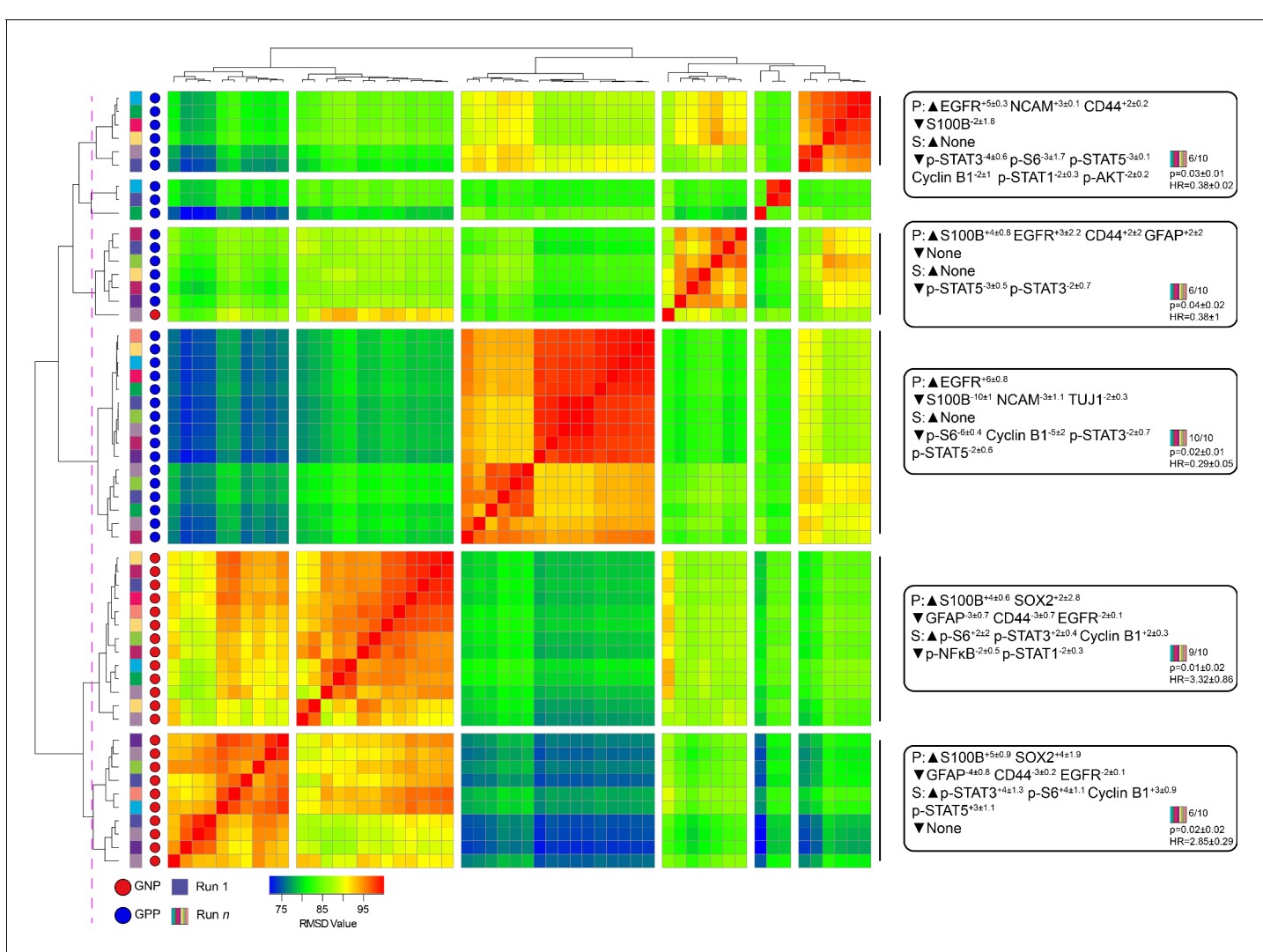

**Figure 3.** Subsampling of glioblastoma cells repeatedly resulted in GNP and GPP subsets with similar phenotypes. RMSD map comparing MEM scores for stable GNP and GPP subsets identified in the main figures and from nine additional t-SNE runs. GNP subsets are noted by red circles and GPP subsets are noted by blue circles. Colored boxes to the left of the red or blue circles indicate the t-SNE run from which the subset is derived. Median MEM labels (± standard deviation) are shown for five major populations to the right. The number of t-SNE analyses represented in each group, as well as median p-value and hazard ratio (HR) are noted in the bottom right corner of each MEM label.

F-measure based on patient categorization (GNP-high, GPP-high, and GNP and GPP low) was 0.79 between t-SNE runs.

To quantify the degree of similarity between the 47 newly identified prognostic clusters and the 8 representative GNP (34, 37, 42) and GPP (2, 3, 4, 5, 41) clusters, the root-mean-square deviation (RMSD) in the MEM enrichment values was calculated (*Diggins et al., 2018*; *Diggins et al., 2017*). GNP subsets from subsequent runs were highly similar to the GNP subsets identified by the initial analysis described above, and the same was observed for GPP subsets (*Figure 3*; GNP v GNP average RMSD = 92.8, GPP v GPP average RMSD = 88.9, and GNP v GPP average RMSD = 80.9). However, some phenotypes were only observed in a small number of t-SNE runs. For example, the phenotype representing cluster 41 was only seen in one other t-SNE. Because this cell type was not observed in at least 50% of the cell sub-samplings, it was considered phenotypically unstable and removed from subsequent analyses (indicated by shading in *Figure 1* and *Figure 1—figure supplement 2*, and asterisks in *Figure 1—figure supplement 3* and *Supplementary file 2*).

## Statistical validation 3: Comparable clusters were identified by RAPID using UMAP instead of t-SNE

To test the modularity of RAPID, the algorithm was implemented using different dimensionality reduction values as input parameters, replacing t-SNE with UMAP, a tool that emphasizes both local and global data structure (*Becht et al., 2019*). RAPID identified 31 populations using UMAP input; 4 of these were prognostic and significantly associated with OS (1 GNP$_{UMAP}$ and 3 GPP$_{UMAP}$) (*Figure 4*). GNP$_{UMAP}$ MEM scores reflected the characteristic S100B and SOX2 co-expression observed in the GNP populations along with an active pro-survival basal signaling status. GPP$_{UMAP}$ subsets were similarly defined by co-expression of EGFR and CD44 and a general lack of the measured phosphorylated signaling effectors (*Figure 4*). When the cells identified using t-SNE were overlaid on the UMAP axes, they occupied similar phenotypic space as UMAP-identified clusters, and vice versa (F-measure for cell assignment to GNP, GPP, or neither = 0.87, *Figure 4*). Thus, when UMAP was used in the RAPID algorithm, GNP and GPP populations were identified that had comparable phenotypes to those identified previously in t-SNE analyses, confirming that RAPID is not dependent upon a specific dimensionality reduction tool (*Figure 4*).

## Statistical validation 4: Risk stratifying cells were continuously associated with outcomes and independent of other glioblastoma stratifying features

At the conclusion of the RAPID analysis, to ensure that results were not an artifact of the high-low cut point choice and to determine if the effect of cell subset abundance was continuous and independent of other features known to stratify glioblastoma survival, a multivariate Cox proportional-hazards model analysis was performed incorporating known predictive features and GNP or GPP cell abundance. The included known predictors were age (*Ohgaki et al., 2004*; *Shapiro et al., 1989*), $O^6$-alkylguanine DNA alkyltransferase (*MGMT*) promoter methylation status (*Brown et al., 2016a*; *Hegi et al., 2005*), and treatment variables including the extent of surgical resection (*Brown et al., 2016b*; *Grabowski et al., 2014*), therapy with temozolomide (*Stupp et al., 2005*), and radiation (*Mirimanoff et al., 2006*; *Walker et al., 1980*). Multivariate survival analysis of GNP cell abundance on a continuous scale, keeping the other predictors constant, indicated that each 1% increase in GNP cells was associated with an approximately 7% increase in mortality compared to baseline (OS HR = 1.07 [95% CI 1.02–1.12], p=0.003). Similarly, a 1% increase in GPP cells was associated with an approximately 7% decrease in mortality rate (OS HR = 0.93 [0.87–1.0], p=0.05) and an approximately 4% increase in time to tumor progression, as compared to baseline (PFS HR = 0.96 [0.93–0.998], p=0.04). When GNP and GPP were assessed simultaneously, abundance of GNP cells was the primary predictor of mortality (OS HR = 1.05 [1.00–1.10], p=0.04), while abundance of GPP cells was the primary predictor of time to tumor progression (PFS HR = 0.96 [0.92–1.00]; p=0.03). Thus, the abundances of GNP and GPP cell subsets were associated with distinct and contrasting patient outcomes (*Figure 1—figure supplement 4*), and their predictive value was independent of each other and known prognostic factors of patient survival.

Since assessing progression-free survival (PFS) can be especially useful in the clinic for cancers with longer median survival, RAPID was also used for the identification of glioblastoma cell clusters

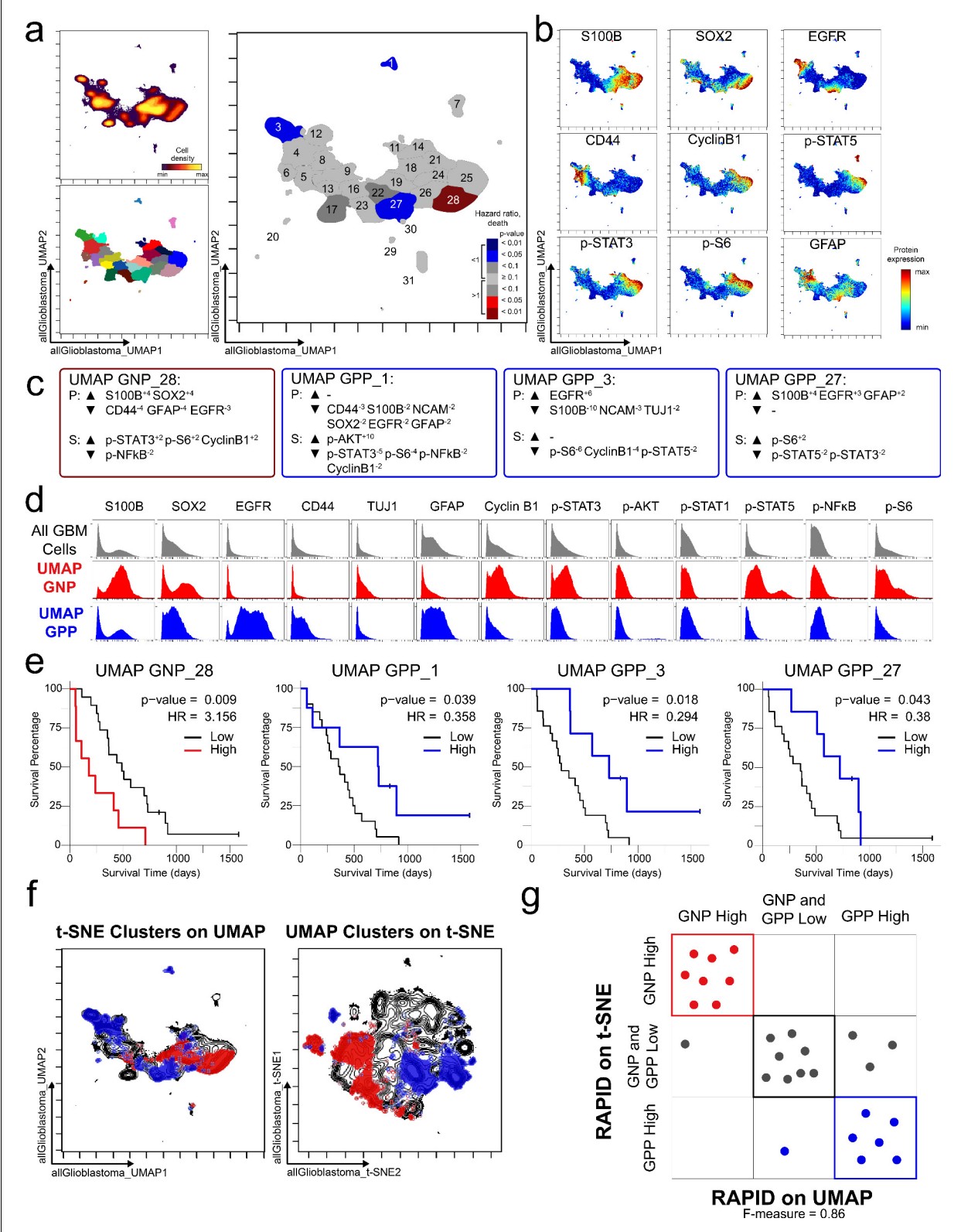

**Figure 4.** GNP and GPP cells were also identified using dimensionality reduction tool UMAP in the RAPID algorithm. (a) UMAP analysis of 131,880 cells from 28 patients. Upper left plot - heat on cell density; lower left plot – colored by FlowSOM cluster; right plot – colored by GNP(red)/GPP(blue) designation and p-value. (b) Per-cell expression levels of 5 identity proteins, 3 phosphorylated signaling effectors, and proliferation marker cyclin B1 are depicted. (c) Enrichment of identity proteins (P) and phosphorylated signaling effectors (S) of glioblastoma cell subsets was quantified using MEM. GNP

*Figure 4 continued on next page*

Figure 4 continued

and GPP cells are labeled in red and blue, respectively. (**d**) Histogram analysis depicts the expression of key identity proteins and phosphorylation signaling effectors of GNP (red) and GPP (blue) compared to all glioblastoma (GBM) cells (gray, top row). (**e**) Overall survival curves for four UMAP-identified populations associated with survival. Cox-proportional hazard model was used to determine a hazard ratio (HR) of death. Censored patients are indicated by vertical ticks. (**f**) GNP (red) and GPP (blue) cells identified via t-SNE ('t-SNE GNP' or 't-SNE GPP') and UMAP ('UMAP GNP' or 'UMAP GPP') are overlaid on either UMAP or t-SNE axes. (**g**) Categorization of each patient (dots) based on GNP high (red), GPP high (blue), or neither (gray) according to abundance based on RAPID using t-SNE or RAPID using UMAP (F-measure = 0.86).

with differential PFS, as opposed to OS. Of the 43 subsets identified by RAPID, 4 subsets were significantly associated with PFS (subsets 20, 33, and 43 with unfavorable PFS ($GNP_{PFS}$) and subset 3 was associated with favorable PFS ($GPP_{PFS}$), *Figure 1—figure supplement 6*).

## Tumors are mosaics of multiple subsets but number of subsets does not correlate with outcome

In the representative t-SNE run (*Figure 1*), RAPID identified 43 phenotypically distinct glioblastoma cell subsets within the tumors analyzed by mass cytometry in this study (*Figure 1*, *Figure 1—figure supplement 4*). The abundance of the 43 clusters varied extensively across patients (*Supplementary file 2*). Tumors contained a median of 14 clusters at >1% with a range from 5 cell clusters in LC06 to a maximum of 27 cell clusters represented in LC25 (*Supplementary file 2*, per-patient maps in *Supplementary file 6*). Although intra-tumor diversity has been hypothesized to contribute to poor response to treatment and survival, here, the number of glioblastoma cell clusters present within a tumor at >1% abundance (a surrogate for intra-tumor diversity) was not observed to be associated with differential survival (ρ = 0.047, p=0.812). In contrast, the abundance of each of the 7 stable and prognostic glioblastoma cell clusters was closely correlated with overall survival (*Figure 1—figure supplement 4*).

## Biological validation 1: A transparent algorithm enables creation of a simple cell identification strategy that captures the cells identified in Dataset 1

After patterns are recognized by a machine learning approach, it is useful to learn from key features and create a straightforward test using alternative technologies or simpler models. One such model is a decision tree using one- or two-dimensional cytometry gating (*Gandelman et al., 2019*), consistent with traditional strategies in immunology and hematopathology. Therefore, a two-dimensional prognostic strategy was designed based on the MEM labels generated from the mass cytometry data. As described above (and *Figure 1—figure supplements 2*, *3* and *4*), MEM labels were generated for each GNP and GPP population, as well as the combined subsets (GNP_Total and GPP_Total), reflecting enriched proteins in each population. These quantitative labels highlighted the most enriched proteins and were used to select S100B (enriched in GNP cells and largely absent from GPP cells) and EGFR (enriched in GPP cells and largely absent from GNP cells) for two-parameter analysis (*Figure 5*). Using only these two proteins, patients could be grouped as GNP-like, GPP-like or GNP and GPP Low, and these groups again exhibited stratified clinical outcomes (HR = 6.56, GNP-like median OS = 111.5 days, GPP-like median OS = 896 days, *Figure 5*). Thus, a simple gating model based on the two most divergent features identified by RAPID was able to meaningfully separate patients into clinically distinct groups.

## Biological validation 2: A larger cohort of glioblastoma samples was stratified using IHC based on phenotypes discovered by RAPID

Unlike fluorescence or mass flow cytometry, IHC is routinely used in surgical pathology. To confirm the ability of S100B and EGFR in separating clinically distinct patient populations using an orthogonal approach, a tissue microarray (TMA) of 73 glioblastoma patient samples was developed. Serial TMA sections were stained with antibodies against S100B and EGFR and the overall signal intensity was determined using QuPath software for each feature (see Methods). By comparing S100B and EGFR staining intensity, patients were scored as GNP-like, GPP-like, or GNP and GPP Low (*Figure 5*). A Kaplan-Meier analysis comparing overall survival between patients enriched with GNP-like cells to those with GPP-like cells confirmed that GNP-like cell enrichment is associated with a shorter overall

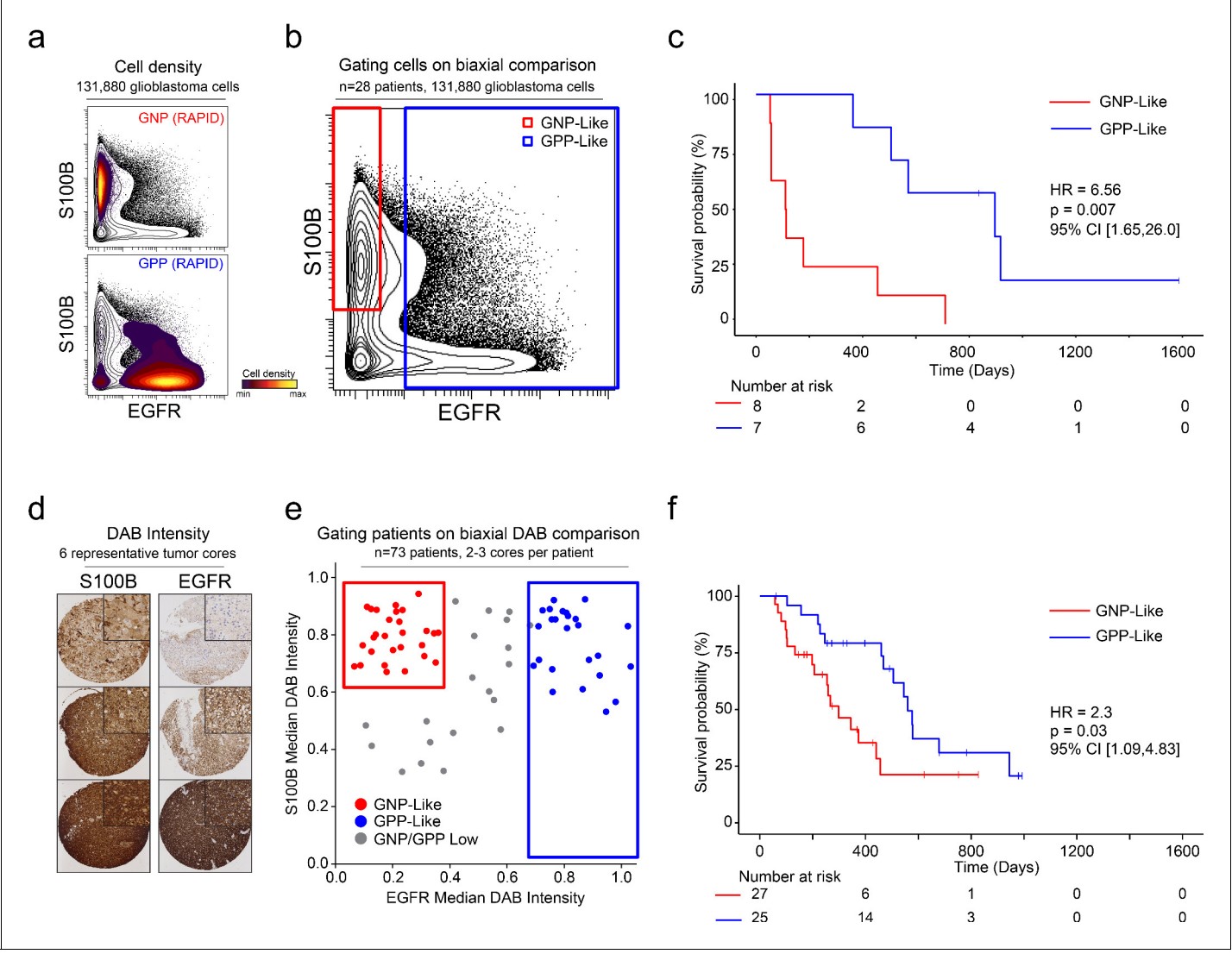

**Figure 5.** A simple gating strategy based on S100B and EGFR can stratify patients using mass cytometry or immunohistochemistry data. (**a**) Biaxial plot of S100B (y-axis) and EGFR (x-axis). Gray contours depict all 131,880 cells from all patients. Density contour overlays depict GNP (top) or GPP (bottom) cells identified by the RAPID algorithm. (**b**) Biaxial plot of S100B (y-axis) and EGFR (x-axis). Gray contours depict all 131,880 cells from all patients as in (a). Red box indicates gate for S100B$^+$/EGFR$^-$ cells, called GNP-like. Blue box indicates gate for EGFR$^+$ cells, called GPP-like. (**c**) Kaplan Meier curve comparing overall survival (in days) of patients with high percentages of GNP-like cells in red (red gate in a, >65.7% = high) and patients with high percentages of GPP-like cells in blue (blue gate in a, >31.2% = high). The hazard ratio of death, calculated using a Cox proportional hazards model, is 6.56 (p=0.0007). (**d**) Example TMA cores stained for S100B (left) or EGFR (right). Brown signal is from 3,3'-Diaminobenzidine (DAB). (**e**) Graph depicting DAB signal intensity for S100B (y-axis) or EGFR (x-axis) from tissue microarray immunohistochemistry on 73 glioblastoma patient samples. The red box outlines patients described as GNP-like (S100B$^{high}$/EGFR$^{low}$) and the blue box outlines patients designated GPP-like (EGFR$^{high}$). All other patients are shown in gray. (**f**) A Kaplan-Meier curve showing overall survival (in days) of patients in the GNP-like (red) or GPP-like (blue) groups. The hazard ratio of death, calculated using a Cox proportional hazards model, is 2.3 (p-value=0.03).

survival (HR = 2.3, GNP-like median OS = 298 days, GPP-like median OS = 560 days, *Figure 5*). These results validated the suspension mass cytometry findings and demonstrated that once revealed by RAPID, GNP-like and GPP-like cells could be identified in new samples by complementary approaches used in laboratory and clinical settings.

## Discussion

The focus of this study was the creation of an unsupervised approach that could work with pilot datasets to suggest prognostic cell types for validation. Ultimately, the RAPID algorithm was tested using numerous statistical approaches, validated with two datasets, and validated as revealing biologically robust cells detectable on other platforms in a larger follow up cohort with formalin-fixed, paraffin-embedded tissue. Prior workflows and algorithms were developed to identify cell populations of interest in cancer samples and emphasized supervised modeling, as with Citrus (*Bruggner et al., 2014*) and Cytofast (*Beyrend et al., 2018*), or comparison to known subsets, as with DDPR (*Good et al., 2018*) and Phenograph (*Levine et al., 2015*). These approaches could not be used with Dataset 1, either because they required a level of prior knowledge about non-malignant adult human brain cells which was not available, or because they required supervision using categorical outcomes, which are not always clearly delineated for continuous variables. Another advantage of RAPID is that it does not require a target cluster number, which is important when it is not known how many phenotypically distinct subsets will be observed in a given cancer type. Cell subsets in tumors can be challenging to manually annotate as they may reasonably be assigned to multiple known cell types, as was apparent here and in prior studies (*Neftel et al., 2019*; *Patel et al., 2014*). RAPID is unsupervised, provides a quantitative label of features enriched in each cluster, and is modular, such that a variety of dimensionality reduction and clustering tools can be used. Currently, a user inputs raw data files (e.g., FCS files from cytometry platforms or equivalent data types from other platforms) and annotated patient survival data. The recommended use of RAPID is to run the full algorithm at least 10 times to seek consensus populations that are stable in phenotype and risk stratification. Both single-run implementation for discovery and a version using these best practices are included as R markdown scripts on the RAPID Github page (https://github.com/cytolab/RAPID). RAPID outputs quantitatively described cell clusters and their significance with respect to patient outcome. While the focus of this study was cytometry data, the design is suitable to other single cell data types where clinical outcomes or similar continuous variables have been scored for pilot cohorts, typically at least 25 individuals. Published datasets were not available for single-cell RNA-seq that matched the criteria for RAPID, including having thousands of cells per sample, more than 25 individuals with annotated clinical outcomes, and multiple features scored consistently for every cell. As single cell RNA-seq and imaging cytometry technologies advance, we anticipate RAPID will be useful for such datasets, especially given how widespread t-SNE, UMAP, and related approaches are within these fields.

The utility of RAPID includes its ability to identify stable, robust clusters that are independent of known prognosticators, provide users with opportunities to customize the workflow with a variety of tools, and inform subsequent studies on validation datasets or using different technologies. Here, RAPID was extensively probed for its performance in each of these areas. By repeated subsampling of each tumor and iterative FlowSOM analyses, clusters with consistent cell content and phenotypes observed in the majority of subsamplings were identified (*Figure 3*). Furthermore, these clusters were independently associated with continuous clinical variables - patient overall survival and PFS. A subsequent, low dimensional decision tree applied to both mass cytometry data and a new set of patient samples stained via IHC was also able to stratify patients, suggesting that the biology learned from the high dimensional approach could be used to inform complementary approaches. Critically, RAPID was also used to analyze a dataset from different tissue in a different disease collected at a different institution, Dataset 2 in *Figure 2* (*Good et al., 2018*). In this application of RAPID, features previously identified by the original authors to be associated with time to relapse were re-captured, identifying cellular phenotypes concordant with prior results without requiring the normal developmental trajectory reference used in the original analysis.

Within Dataset 1 analyzing 28 *IDH* wild-type pre-therapy glioblastoma patient samples, the RAPID workflow automatically uncovered two prognostic phenotypic signatures which were independent of other known predictors of outcome. Glioblastoma Negative Prognostic (GNP) cells, characterized by enrichment for S100B, SOX2, p-STAT3, and p-STAT5, were associated with decreased overall survival, while Glioblastoma Positive Prognostic (GPP) cells, characterized by co-enrichment of EGFR and CD44 proteins, were associated with longer overall survival. Once revealed in high-dimensional data, a simple gating scheme using S100B and EGFR could be used to stratify outcome in a separate, expanded set of samples using traditional pathological approaches. High-dimensional

cytometry and RAPID were critical to revealing novel prognostic cells in glioblastoma data in two ways. First, assessment of a large number of cells per tumor – over 2 million viable single cells, with at least 4,710 glioblastoma cells from each patient - enabled the use of an unsupervised approach in the identification of rare, novel cell subsets across patients. Second, per-cell quantification of phosphorylated signaling effector proteins revealed potential mechanisms of tumor cell regulation that are not readily apparent in bulk tumor data, genomic analyses, or lower dimensional approaches such as one- to four-color imaging. Supervised analysis of single cell data has previously uncovered signaling events tied to patient survival in hematologic malignancies (*Good et al., 2018*; *Irish et al., 2004*; *Levine et al., 2015*; *Myklebust et al., 2017*), and a similar pattern was observed here.

The GNP signature was defined by abnormal neural development features such as co-expression of stem cell transcription factor SOX2 and astrocyte lineage marker S100B (*Ikushima et al., 2009*; *Raponi et al., 2007*) and simultaneous high basal phosphorylation of multiple signaling effectors downstream of receptor tyrosine kinases reported to be important in tumor biology (*Bhat et al., 2013*; *Carro et al., 2010*; *Dolma et al., 2016*; *Fan et al., 2017*; *Tan et al., 2019*; *Wei et al., 2013*; *Figure 1—figure supplement 4*). RAPID also uncovered a connection between p-STAT5 and glioblastoma outcome previously unidentified in primary patient samples. STAT5 signaling is required in development of many tissues to block apoptosis and drive cell cycle entry (*Irish et al., 2006*) for example, p-STAT5 is an essential feature of negative prognostic acute myeloid leukemia signaling profiles (*Irish et al., 2004*; *Levine et al., 2015*). The signaling events of the negative and positive prognostic cells can now be studied in glioblastoma research models, such as patient xenografts and glioblastoma organoids (*Bhaduri et al., 2020*; *Hubert et al., 2016*; *Jacob et al., 2020*; *Ogawa et al., 2018*), using new combinations of targeted therapies, such as JAK inhibitors that target molecules upstream of STAT5 and STAT3, in combination with PI3K/mTOR pathway inhibitors, which will target molecules upstream of AKT and S6 signaling. In this way, new combinations of existing therapies may prove useful in targeting the signaling that defines the negative prognostic cells seen here.

Recent work using single cell gene expression has described the existence of multiple cellular states in glioblastoma tumors and the ability of cells to transition between states (*Neftel et al., 2019*). Similar to most transcript-based studies, RAPID analyses were performed on cells collected at a single timepoint, precluding a direct investigation of the ability of GNP or GPP cells to transition to other phenotypes; however, it is possible that phosphorylated, active STAT3, STAT5, and S6 may enable transition between progenitor-like states as they do in earlier development, and thus influence patient outcome (*Rushing et al., 2019*; *Yoshimatsu et al., 2006*). Another key research question for the future will be whether the signature features of the risk stratifying cells seen here will also be seen in other types of intractable human malignancies. Intriguingly, p-STAT5, p-ERK, and p-STAT3 signaling profiles reminiscent of the negative prognostic cells from glioblastoma have been seen in leukemia (*Irish et al., 2004*; *Kotecha et al., 2008*; *Levine et al., 2015*) and ovarian cancer (*Gonzalez et al., 2018*).

The GPP signature, in contrast, was defined by EGFR and CD44 co-enrichment, diminished evidence of proliferation, and specific lack of STAT5 phosphorylation. GPP cells were further associated with higher proportions of tumor-infiltrating immune cells. This result suggests an understanding of prognostic cell content or biomarkers may be relevant for immunotherapy research in glioblastoma. Previous DNA and RNA-driven molecular subtyping predicts EGFR expression in the classical subset of glioblastoma tumors and CD44 expression in mesenchymal tumors (*Verhaak et al., 2010*). As these categories were primarily based on bulk tumor data, cells co-expressing EGFR and CD44 (classified as GPP cells in this study) may have previously been missed, although single glioma cells have been shown to simultaneously amplify sequence or co-express transcripts for important signaling regulators (*Patel et al., 2014*; *Snuderl et al., 2011*). EGFR has been extensively studied as a driver of gliomas in the past (reviewed in *Saadeh et al., 2018*), and the association of this gene and transcript with outcome has been a matter of debate (*Li et al., 2018*; *Saadeh et al., 2018*; *Xu et al., 2017*). This study finds that expression of EGFR protein is associated with better overall survival. One reason for the difference between this study and other reports may be that EGFR protein levels were measured in individual cells rather than copy number analysis or transcript levels in bulk tumor samples; our own analyses and others' have indicated that copy number or transcript level are not necessarily predictive of protein expression (*Baser et al., 2019*; *Brennan et al., 2009*; *Chakravarty et al., 2017*). Although antibody-based methods for protein detection, like those used

here, depend on the specificity of each selected clone, it is important to note that two different, rigorously validated antibodies (mass cytometry, clone AY13; TMA, clone A-10) gave the same results (*Figure 5*). S100B has been explored as a serum biomarker (*Holla et al., 2016*), and S100B is known for its impact on macrophages, including microglia (*Wang et al., 2013*). These features of negative and positive prognostic cells extend the single cell phospho-specific flow cytometry approach to a new solid tumor that is in urgent need of new biological insights and targets.

When applied to a new glioblastoma dataset as well as a previously published study of blood cancer, RAPID reliably identified cells whose abundance was predictive of good or poor outcome. Cellular identification was robust, stable, and reproducible, and independent of the specific dimensionality reduction tools used. Critically, the discoveries from RAPID were able to inform a scoring system for detection of GNP-like and GPP-like phenotypes in IHC data that stratified patient outcome in 73 patient samples. RAPID also led to the development of a lower-dimensional cytometry pipeline which could be optimized for clinical stratification. There is now the exciting potential to extend the hypotheses suggested by RAPID into clinical research studies using either traditional flow cytometry or IHC on widely available formalin-fixed, paraffin-embedded samples, as in the biological validation here (*Figure 5*). Thus, techniques accessible to clinical research, such as IHC, could be informed by the results from RAPID and envisioned as a way to assign glioblastoma patients to treatment groups in early phase clinical trials.

# Materials and methods

## Key resources table

| Reagent type (species) or resource | Designation | Source or reference | Identifiers | Additional information |
|---|---|---|---|---|
| Biological sample (*Homo Sapien*) | Primary glioblastoma tumors | Vanderbilt University Medical Center | | Freshly isolated from primary glioblastoma resections |
| Reagent | Rhodium | Fluidigm | Cat# 201103A | MC (1:4000) |
| Antibody | Anti-Cyclin B1 (mouse-monoclonal) | BD Biosciences | RRID:AB_395287 Cat#554176 Clone: GNS-1 | MC (1:100) |
| Antibody | Anti-TUJ1 (mouse-monoclonal) | Biolegend | RRID:AB_2313773 Cat#801201 Clone: TUJ1 | MC (1:100) |
| Antibody | Anti-cCasp3 (rabbit-monoclonal) | Fluidigm | RRID:AB_2847863 Cat#3142004A Clone: 5A1E | MC (1:100) |
| Antibody | Anti-CD117 (mouse-monoclonal) | Fluidigm | RRID:AB_2847864 Cat#3143001B Clone:104D2 | MC (1:100) |
| Antibody | Anti-S100B (mouse-monoclonal) | BD Biosciences | RRID:AB_647296 Cat#612376 Clone: 19/S100B | MC (1:100) |
| Antibody | Anti-CD31 (mouse-monoclonal) | Fluidigm | RRID:AB_2737262 Cat#3145004B Clone: WM59 | MC (1:100) |
| Antibody | Anti-γH2AX (mouse-monoclonal) | Fluidigm | RRID:AB_2847865 Cat# 3147016A Clone: JBW301 | MC (1:100) |
| Antibody | Anti-CD34 (mouse-monoclonal) | Fluidigm | RRID:AB_2810243 Cat#3148001B Clone: 581 | MC (1:100) |
| Antibody | p-4E-BP1 (T37/T46) | Fluidigm | RRID:AB_2847866 Cat# 3149005A Clone: 236B4 | MC (1:100) |

*Continued on next page*

*Continued*

| Reagent type (species) or resource | Designation | Source or reference | Identifiers | Additional information |
|---|---|---|---|---|
| Antibody | Anti-p-STAT5 (Y694) (mouse-monoclonal) | Fluidigm | RRID:AB_2744690 Cat#3150005A Clone:47 | MC (1:100) |
| Antibody | Anti-BMX (mouse-monoclonal) | BD Biosciences | RRID:AB_2290762 Cat# 610793 Clone: 40/BMX | MC (1:100) |
| Antibody | Anti-p-AKT (S473) (rabbit-monoclonal) | Fluidigm | RRID:AB_2811246 Cat#3152005A Clone: D9E | MC (1:100) |
| Antibody | Anti-p-STAT1 (Y701) (rabbit-monoclonal) | Fluidigm | RRID:AB_2811248 Cat#3153003A Clone: 58D6 | MC (1:100) |
| Antibody | Anti-CD45 (mouse-monoclonal) | Fluidigm | RRID:AB_2810854 Cat# 3154001B Clone: HI30 | MC (1:400) |
| Antibody | Anti-NCAM/CD56 (mouse-monoclonal) | Biolegend | RRID:AB_604092 Cat# 318302 Clone: HCD56 | MC (1:100) |
| Antibody | Anti-p-p38 (T180/Y182) (rabbit-monoclonal) | Fluidigm | RRID:AB_2661826 Cat# 3156002A Clone: D3F9 | MC (1:100) |
| Antibody | Anti-p-STAT3 (Y705) (mouse-monoclonal) | Fluidigm | RRID:AB_2811100 Cat#3158005A Clone: 4/P-STAT3 | MC (1:100) |
| Antibody | Anti-ITGα6/CD49F (rat-monoclonal) | Biolegend | RRID:AB_345296 Cat# 313602 Clone: GoH3 | MC (1:100) |
| Antibody | Anti-CD133 (mouse-monoclonal) | Miltenyi Biotech | RRID:AB_244339 Cat# 130-090-422 Clone: AC133 | MC (1:50) |
| Antibody | Anti-PDGFRα (mouse-monoclonal) | Biolegend | RRID:AB_755996 Cat#323502 Clone: 16A1 | MC (1:50) |
| Antibody | Anti-SOX2 (mouse-monoclonal) | BD Biosciences | RRID:AB_10694256 Cat# 561469 Clone: O30-678 | MC (1:100) |
| Antibody | Anti-SSEA-1/CD15 (mouse-monoclonal) | Fluidigm | RRID:AB_2810970 Cat# 3164001B Clone: W6D3 | MC (1:100) |
| Antibody | Anti-EGFR (mouse-monoclonal) | Biolegend | RRID:AB_10945161 Cat# 352902 Clone:AY13 | MC (1:100) |
| Antibody | Anti-p-NFκB p65 (S529) (mouse-monoclonal) | Fluidigm | RRID:AB_2847867 Cat# 3166006A Clone: K10-895.12.50 | MC (1:100) |
| Antibody | Anti-L1CAM (mouse-monoclonal) | BD Biosciences | RRID:AB_395337 Cat#554273 Clone: 5G3 | MC (1:100) |
| Antibody | Anti-Nestin (mouse-monoclonal) | Millipore | RRID:AB_2251134 Cat# MAB5326 Clone:10C2 | MC (1:100) |
| Antibody | Anti-CD44 (mouse-monoclonal) | Biolegend | RRID:AB_1501199 Cat# 338802 Clone: BJ18 | MC (1:100) |

*Continued*

| Reagent type (species) or resource | Designation | Source or reference | Identifiers | Additional information |
|---|---|---|---|---|
| Antibody | Anti-GFAP (mouse-monoclonal) | BD Biosciences | RRID:AB_396366 Cat# 556328 Clone: 1B4 | MC (1:200) |
| Antibody | Anti-p-ERK1/2 (T202/Y204) (rabbit-monoclonal) | Fluidigm | RRID:AB_2811250 Cat#3171010A Clone: D13.14.4E | MC (1:100) |
| Antibody | Anti-p-S6 (S235/S236) (mouse-monoclonal) | Fluidigm | RRID:AB_2811251 Cat#3172008A Clone: N7-548 | MC (1:100) |
| Antibody | Anti SOX10 (mouse-monoclonal) | Santa Cruz | RRID:AB_10844002 Cat#sc-365692 Clone: A-2 | MC (1:100) |
| Antibody | Anti-HLA-DR (mouse-monoclonal) | Fluidigm | RRID:AB_2665397 Cat# 3174001B Clone: L243 | MC (1:200) |
| Antibody | Anti-p-HH3 (rat-monoclonal) | Fluidigm | RRID:AB_2847869 Cat# 3175012A Clone: HTA28 | MC (1:400) |
| Antibody | Anti-Histone H3 (rabbit-monoclonal) | Fluidigm | RRID:AB_2847870 Cat# 3176016A Clone: D1H2 | MC (1:200) |
| Antibody | S100B (rabbit-polyclonal) | Dako | RRID:AB_2811056 Cat#GA50461-2 | IHC (RTU) |
| Antibody | EGFR | Santa Cruz | RRID:AB_10920395 Cat# sc-373746 Clone: A-10 | IHC (1:100) |
| Software, algorithm | RAPID | https://github.com/cytolab/RAPID | | |
| Data files | FCS data files | https://flowrepository.org/id/FR-FCM-Z24K | | |

## Lead contact and materials availability

Further information and requests for datasets and materials should be addressed to jonathan.irish@vanderbilt.edu.

## Experimental model and subject details

### Patient samples

Surgical resection specimens of 28 *IDH*-wildtype glioblastomas collected at Vanderbilt University Medical Center between 2014 and 2016 were processed into single cell suspensions following an established protocol (*Leelatian et al., 2017b*). Only samples that were confirmed to be *IDH*-wildtype glioblastomas by standard pathological diagnosis were used. All samples were collected with patient informed consent in compliance with the Vanderbilt Institutional Review Board (IRBs #030372, #131870, #181970), and in accordance with the declaration of Helsinki.

### Patient characteristics and collection of clinical data

Additional patient characteristics are included in *Supplementary file 3* for all samples in this study. All patients were adults (≥18 years of age) at the time of their maximal safe surgical resection of their cerebral (supratentorial) glioblastomas. Extent of surgical resection was independently classified as either gross total or subtotal resection by a neurosurgeon and a neuroradiologist. Gross total resection was defined as agreement by both viewers of no significant residual tumor enhancement on patients' gadolinium-enhanced magnetic resonance imaging (MRI) of the brain obtained within

24 hr after surgery. All patients were considered for treatment with postoperative chemotherapy (temozolomide) and radiation according to the standard of care (*Stupp et al., 2005*), after determination of *MGMT* promoter methylation status by pyrosequencing (Cancer Genetics, Inc, Los Angeles, CA, USA). Multiplex polymerase chain reaction (PCR) was used to determine *IDH1/2* mutational status. Patients' postoperative course was followed until February 2019, noting time to first, definitive radiographic progression or recurrence of glioblastoma as agreed upon by the treating neuro-oncologist and neuroradiologist, and the time to patients' death. All deaths were deemed to be due to the natural course of patients' glioblastoma. Median overall survival of the analyzed 28 patients with *IDH* wild-type glioblastoma was 388.5 days (13 months) and median PFS was 187.5 days (6.3 months), which is typical for the disease (*Ostrom et al., 2017*; *Stupp et al., 2005*).

## Method details
### Mass cytometry analysis
Cells derived from patient samples were prepared as previously described (*Leelatian et al., 2017b*). A multi-step staining protocol was used, which included 1) live surface stain, 2) 0.02% saponin permeabilization intracellular stain, and 3) intracellular stain after permeabilization with ice-cold methanol. All antibodies used, including clone information, and the steps when used are given in *Supplementary file 4*. After staining, cells were resuspended in deionized water containing standard normalization beads (Fluidigm) (*Finck et al., 2013*), and collected on a CyTOF 1.0 instrument located in the Cancer and Immunology Core facility at Vanderbilt University. Mass cytometry standardization beads were used to remove batch effects and to set the variance stabilizing arcsinh scale transformation for each channel following field-standard protocols (*Greenplate et al., 2019*; *Leelatian et al., 2015*; *Leelatian et al., 2017b*). Rhodium viability stain and cleaved caspase-3 antibody were included in staining to exclude non-viable and apoptotic cells, respectively. Detection of total histone H3 was used to identify intact, nucleated cells (*Leelatian et al., 2017a*). A 34-dimensional mass cytometry antibody panel was used to analyze over 2 million viable cells from 28 tumors (ranging from 4860 to 336,284 cells per tumor). Data were normalized with MATLAB-based normalization software (*Finck et al., 2013*), and were arcsinh transformed (cofactor 5), prior to analysis using the Cytobank platform (*Kotecha et al., 2010*). Positively identified cells were defined by having signal above 10 on any channel on which an antibody was used to detect antigen. A patient-specific t-SNE view was generated, using 26 of the measured markers for all tumor and stromal cells from each patient's tumor (*Amir et al., 2013*; *Supplementary file 4*). Immune (CD45$^+$) and endothelial cells (CD31$^+$) were computationally excluded from each individual patient prior to subsequent downstream analysis. Remaining CD45$^-$CD31$^-$ cells were included in a common t-SNE analysis, generated using 24 of 34 measured markers (*Supplementary file 4*). Distribution of each of the 28 patients' cells on the common t-SNE axes and mass intensity for each marker are shown in *Supplementary file 6*. This common t-SNE analysis was used for automated analysis of risk stratifying cell subsets in RAPID (below).

## Quantification and statistical analysis
### Implementation of RAPID in R
FCS files for each patient sample (28) containing only cells of interest (non-immune, non-endothelial cells) were input in R (4,710 cells from each patient, 131,880 cells total). Cell subset identification was performed using the previously published FlowSOM R package (*Van Gassen et al., 2015*). t-SNE values (t-SNE1_glioblastoma and t-SNE2_glioblastoma) from t-SNE (or UMAP values from UMAP) analysis of CD45$^-$CD31$^-$ glioblastoma cells from 28 patients were used as parameters for cell subset clustering. Within the RAPID workflow, the optimal number of clusters was determined by first identifying, for each feature, the smallest number of clusters that minimizes the intra-cluster signal variance for that feature. Then, the optimal cluster number of the data set was determined by taking the median of the optimal numbers for each individual feature. Once the cluster number was determined, the abundance of cell subsets and their clinical significance was assessed using outcome-guided analysis. Patients were divided into Low and High groups, based on the distribution (interquartile variance, IQR) of the abundance of a given cell subset across the cohort. A univariate Cox regression analysis was then used to estimate the effect size (hazard ratio, HR, of death) on survival and quantify its statistical significance with a p-value. The RAPID program output included: 1) a

PDF containing two color coded, 2D t-SNE (or UMAP) plots (.png), one depicting all FlowSOM clusters and one depicting prognostic status and p-value, Kaplan-Meier survival plots of patients for each subset; 2) MEM outputs including a PDF of the MEM heatmap as well as. txt files of MEM and Median values for each feature, enrichment scores, and IQR values; 3) a .txt file of the FlowSOM cluster value for prognostic subsets, a .txt file of survival statistics for each FlowSOM cluster, and a .csv file with subset abundance information per patient;and 4) new FCS files with added columns for cluster and prognostic status for each cell. In this study, abundance of Glioblastoma Negative Prognostic (GNP) and Glioblastoma Positive Prognostic (GPP) cells in each tumor was quantified as percentages per total glioblastoma cells (i.e. immune and endothelial cells were already excluded). Total GNP and GPP cell abundance was determined for each patient by adding the events in all GNP (or GPP subsets, respectively) together. GNP high patients were identified as containing more GNP cells than the IQR of total GNP abundance (3.1%). GPP high patients were defined in the same manner (total GPP cell abundance IQR = 8.58%). MEM analysis was performed in R, using the previously published R package (*Diggins et al., 2017*). In short, MEM captured and quantified cell subset-specific feature enrichment by scaling the magnitude (median) differences between clusters, depending on the spread (IQR) of the data. These values were then computed in comparison to the remaining cells in a given dataset. MEM values were interpreted as either being positively enriched (▲, UP positive values) or negatively enriched (▼, DN negative values). The variation of a given cellular feature across GNP or GPP cell subsets was quantified as ± standard deviations (SD). For the primary data set used in this study (131,880 cells), RAPID ran in 15 min from start to finish after dimensionality reduction.

## Cluster stability testing

Ten independent t-SNE analyses were performed on equal numbers of randomly sampled cells from each patient (4,710 cells per patient, 131,880 total cells). RAPID was used to analyze each of these ten t-SNE runs. For each sub-sampling of cells and the respective t-SNE, an additional 99 FlowSOM clusterings were performed without setting a seed for reproducible results. After each analysis, an F-measure was calculated per cluster, measuring both the precision and recall of cell assignment. After 100 total FlowSOM runs, each of the original clusters had an average F-measure, interpreted here as a measure of cluster stability.

## Survival and statistical analysis

Time from surgical resection to death (overall survival, OS) and time from surgical resection to the initial radiographic recurrence or death before radiographic assessment (PFS) were depicted using right-censored Kaplan-Meier curves and analyzed in R. Survival time points were censored if, at last follow up, the patient was known to be alive or had not had radiographic progression. Differences in the survival curves of groups were compared using the Cox univariate regression model, reporting a hazard ratio (HR) with 95% confidence intervals between the survival curves.

A Cox proportional-hazards regression model was created to assess the influence of GNP and GPP cells on OS and PFS as continuous variables while accounting for other factors known to affect survival, including age at diagnosis, *MGMT* promoter methylation status, extent of surgical resection (EOR), treatment with temozolomide (TMZ), and radiation (XRT). The hazard model can be written as:

$$HR = \frac{h(t)}{h_0(t)} = e^{\left(b_{GNP}GNP + b_{age}Age + b_{MGMT}MGMT + b_{EOR}EOR + b_{XRT}XRT + b_{TMZ}TMZ\right)}$$

where $\frac{h(t)}{h_0(t)}$ represents the ratio of hazard comparing the risk of death at time $t$ to the baseline hazard (obtained when all variables are equal to zero) and $e^{b_x}$ represents the hazard ratio of variable $x$. The data were fit using R software, version 3.5 (R foundation for Statistical Computing, Vienna, Austria). The proportional-hazards assumption was tested in all multivariate models and supported by a non-significant relationship between Schoenfeld residuals and time for each covariate included in the model (p > 0.38; degree of freedom = 1) and the overall model (p = 0.96; degrees of freedom = 6 and 7). Statistical significance α was set at 0.05 for all statistical analyses, one- or two-tailed noted in figure legends.

An F-measure was used to quantify the level of agreement between classifications of patients or cells between alternative analysis strategies as wells as multiple RAPID iterations. The F-measure is the harmonic mean of the precision and recall given by the equation F = 2 * (Precision * Recall) / (Precision + Recall) where Precision = True Positive / (True Positive + False Positive) and Recall = True Positive / (True Positive + False Negative). An F-measure of 1 indicates perfect agreement between two different strategies or iterations as opposed to an F-measure of 0 which would mean no agreement between classifications of patients or cells from two strategies or iterations. Patients could be classified as GNP high, GNP and GPP low, or GPP high, while cells were classified as GNP, GPP, or neither. None of the patients in this study were classified as both GNP high and GPP high. To calculate the F-measure of patient categorization, the classification of the 28 patients into the three prognostic groups from the t-SNE implementation of RAPID was used as the reference point from which to compare patient classification resulting from the UMAP implementation of RAPID. Similarly, the stability of the RAPID workflow in assigning cells to GNP, GPP, or non-significant clusters was tested by using the t-SNE implementation of RAPID (FlowSOM seed 38) as the reference from which to compare 100 iterations of RAPID (random FlowSOM seed per iteration). Calculation of the F-measure was implemented using R software, version 3.5.

## Computer specifications

R was downloaded from https://cran.r-project.org/bin/ and implemented using the R Studio GUI https://www.rstudio.com/products/rstudio/download/#download. PC users also needed to download R Tools https://cran.r-project.org/bin/windows/Rtools/ and MAC users needed to download X11 Quartz https://www.xquartz.org/. RAPID was implemented, using these tools, on several personal computers. It was developed on a Dell Precision 7820 with a solid state hard drive and 64 GB RAM.

## Tissue microarray construction and analysis

### TMA sample selection

Formalin-fixed paraffin-embedded (FFPE) glioblastoma specimens were identified using the Vanderbilt Surgical Pathology database. The absence of *IDH* mutation was determined by multiplex PCR coupled with base extension assay (SNaPshot reaction mixture, Life Technologies, Carlsad, CA, USA), followed by capillary electrophoresis on an ABI Genetic Analyzer 3130XL and GeneMapper v.4.1. Following confirmation of the previously rendered histologic diagnosis, hematoxylin and eosin stained slides were scanned on the Panoramic P250 (3DHistech) whole slide scanner. Areas containing viable tumor were identified and circled by two pathologists (BM, NL).

### TMA construction and staining

Blocks were delivered to the Vanderbilt University Medical Center TPSR (Translational Pathology Shared Resource), where cores were extracted from the encircled areas. Donor blocks and recipient blocks were loaded into the Tissue Microarray Grandmaster (3DHistech). The virtual slide images were aligned and overlaid on the tissue block and cores were removed from the donor block based on the pathologist annotation. Three 1 mm core samples were collected from each tumor and placed in the recipient block. IHC of serial sections of two TMA blocks (<10 µm thick) were stained with primary antibodies conjugated to HRP and 3,3′-Diaminobenzidine (DAB) detection for EGFR and S100B, and counter stained with Hematoxylin by the Translational Pathology Shared Resource (TPSR) at Vanderbilt University. Digital images were obtained with an Ariol SL-50 automated scanning microscope and the Leica SCN400 Slide Scanner from VUMC Digital Histology Shared Resource.

| Marker | Clone | Company |
| --- | --- | --- |
| S100B | polyclonal | Dako |
| EGFR | A-10 | Santa Cruz Biotechnology |

## TMA imaging and analysis

Whole slide imaging was performed in the Digital Histology Shared Resource at Vanderbilt University Medical Center (www.mc.vanderbilt.edu/dhsr). For each marker, a QuPath project was created and all slide images were uploaded to be processed in batch. In QuPath, regions of interest (ROI's) were designated by circling each tumor core. Each ROI was computationally linked to the patient by a unique identifier, allowing cores from the same patient to be grouped. For each marker, the 'Estimate Stain Vectors' function in QuPath was used to find the appropriate deconvolution parameters to isolate the signal intensity contribution from Hematoxylin and DAB respectively. The deconvolution parameters are listed below:

| Marker | Hematoxylin | | | DAB | | | Background | | |
|---|---|---|---|---|---|---|---|---|---|
| S100B | 0.60484 | 0.67532 | 0.422044 | 0.20996 | 0.50234 | 0.83879 | 224 | 223 | 221 |
| EGFR | 0.72353 | 0.63737 | 0.26508 | 0.24952 | 0.52384 | 0.81445 | 221 | 219 | 220 |

For each ROI, nuclear segmentation on the Hematoxylin Optical Density (OD) was optimized using the 'Watershed cell detection' function in QuPath, and the cytoplasm around each nucleus was estimated by performing a 3 μm expansion from the nuclear outline. All measurements from all detections were exported for analysis in R. In R, specific parameters (Name, Cell.DAB.OD.mean, Cytoplasm.DAB.OD.mean, and Nucleus.DAB.OD.mean) were extracted for every detection (cell) from every patient. These parameters identify the ROI/core from which the cell was segmented, its corresponding patient ID, the mean optical density of the deconvoluted DAB signal in each entire segmented cell, the DAB signal in only the cytoplasm, and the signal exclusively in the nucleus respectively. The full TMA map linking QuPath IDs, Patient_IDs, Block, and Core_IDs was also imported. In addition, for each marker, the median DAB intensity was calculated for each patient (averaged over three cores). The thresholds and measurements on which these thresholds were applied are summarized below:

| Marker | Measurement | Threshold - Block A | Threshold - Block B |
|---|---|---|---|
| S100B | Cell_DAB | 0.4 | 0.4 |
| EGFR | Cell_DAB | 0.2 | 0.2 |

Patients were categorized as GNP-like if their TMA cores had S100B staining intensity above the first quartile of S100B intensities (>0.6728) and had EGFR staining below the 50th percentile (<0.4199). Patients were categorized as GPP-like if their TMA cores scored in the top tertile of EGFR intensity (>0.6929).

## Data and code availability

### Data availability

Annotated flow data files are available at the following link https://flowrepository.org/id/FR-FCM-Z24K. FCS files that contain the cells from the representative t-SNE can also be found on the GitHub page: https://github.com/cytolab/RAPID. Patient-specific views of population abundance and channel mass signals for all analyzed patients in this study are found in *Supplementary file 6*.

### Code availability

RAPID code is currently available on Github, along with FCS files from Dataset 1 and 2 for analysis, at: https://github.com/cytolab/RAPID '2020-01-15 RAPID Workflow Script on Davis Dataset.Rmd' contains RAPID code for a single run as presented in *Figure 1b*. '2020-04-21 RAPID Stability Tests. Rmd' contains RAPID code for repeated stability tests as presented in *Figure 1c*.

## Acknowledgements

We thank the Irish and Ihrie labs at Vanderbilt University for helpful discussions.

Research was supported by the following funding resources: NIH/NCI R00 CA143231 (JMI), the Vanderbilt-Ingram Cancer Center (VICC, P30 CA68485), the Vanderbilt International Scholars Program (NL), a Vanderbilt University Discovery Grant (JMI and NL), Alpha Omega Alpha Postgraduate Award (AMM), Society of Neurological Surgeons/RUNN Award (AMM), F32 CA224962-01 (AMM), 2018 Burroughs Wellcome Fund Physician-Scientist Institutional Award 1018894 (AMM), T32 HD007502 (JS), F31 CA199993 (ARG), R25 CA136440-04 (KED), a VICC Provocative Question award (JMI), R01 CA226833 (JMI), U54 CA217450 (JMI), U01 AI125056 (JMI and SMB.), R01 NS096238 (RAI), DOD W81XWH-16-1-0171 (RAI), the Michael David Greene Brain Cancer Fund (RAI), the Vanderbilt Institute for Clinical and Translational Research (VR51342, RAI, BCM), VICC Ambassadors awards (JMI and RAI), and the Southeastern Brain Tumor Foundation (JMI and RAI).

## Additional information

### Competing interests

Jonathan M Irish: was a co-founder and a board member of Cytobank Inc and received research support from Incyte Corp, Janssen, and Pharmacyclics. The other authors declare that no competing interests exist.

### Funding

| Funder | Grant reference number | Author |
|---|---|---|
| National Institutes of Health | R00 CA143231 | Jonathan M Irish |
| Vanderbilt Ingram Cancer Center | P30 CA68485 | Jonathan M Irish |
| Vanderbilt University | International Scholars Program | Nalin Leelatian |
| Vanderbilt University | Discovery Grant | Nalin Leelatian Jonathan M Irish |
| Alpha Omega Alpha Honor Medical Society | Postgraduate Award | Akshitkumar M Mistry |
| Society of Neurological Surgeons | RUNN Award | Akshitkumar M Mistry |
| National Institutes of Health | F32 CA224962-01 | Akshitkumar M Mistry |
| Burroughs Wellcome Fund | 1018894 | Akshitkumar M Mistry |
| National Institutes of Health | T32 HD007502 | Justine Sinnaeve |
| National Institutes of Health | F31 CA199993 | Allison R Greenplate |
| National Institutes of Health | R25 CA136440-04 | Kirsten E Diggins |
| Vanderbilt Ingram Cancer Center | Provocative Question | Jonathan M Irish |
| National Institutes of Health | R01 CA226833 | Jonathan M Irish |
| National Institutes of Health | U54 CA217450 | Jonathan M Irish |
| National Institutes of Health | U01 AI125056 | Sierra M Barone Jonathan M Irish |
| National Institutes of Health | R01 NS096238 | Rebecca A Ihrie |
| U.S. Department of Defense | W81XWH-16-1-0171 | Rebecca A Ihrie |
| Michael David Greene Brain Cancer Fund | | Rebecca A Ihrie Jonathan M Irish |
| Vanderbilt Institute for Clinical and Translational Research | VR51342 | Bret C Mobley Rebecca A Ihrie |
| Vanderbilt Ingram Cancer Center | Ambassadors Award | Rebecca A Ihrie Jonathan M Irish |

Southeastern Brain Tumor
Foundation

Rebecca A Ihrie
Jonathan M Irish

The funders had no role in study design, data collection and interpretation, or the decision to submit the work for publication.

### Author contributions
Nalin Leelatian, Justine Sinnaeve, Conceptualization, Data curation, Formal analysis, Supervision, Validation, Investigation, Visualization, Methodology, Writing - original draft, Project administration, Writing - review and editing; Akshitkumar M Mistry, Resources, Software, Formal analysis, Funding acquisition, Validation, Methodology, Writing - review and editing; Sierra M Barone, Data curation, Software, Formal analysis, Validation, Visualization, Methodology, Writing - review and editing; Asa A Brockman, Data curation, Software, Formal analysis, Validation; Kirsten E Diggins, Software; Allison R Greenplate, Software, Writing - review and editing; Kyle D Weaver, Reid C Thompson, Lola B Chambless, Resources; Bret C Mobley, Conceptualization, Resources, Formal analysis, Supervision, Validation, Methodology, Writing - review and editing; Rebecca A Ihrie, Conceptualization, Resources, Supervision, Funding acquisition, Methodology, Project administration, Writing - review and editing; Jonathan M Irish, Conceptualization, Resources, Software, Supervision, Funding acquisition, Methodology, Project administration, Writing - review and editing

### Author ORCIDs
Justine Sinnaeve (iD) http://orcid.org/0000-0001-9303-7969
Akshitkumar M Mistry (iD) http://orcid.org/0000-0002-7918-5153
Sierra M Barone (iD) http://orcid.org/0000-0001-5944-750X
Rebecca A Ihrie (iD) https://orcid.org/0000-0003-0439-0141
Jonathan M Irish (iD) https://orcid.org/0000-0001-9428-8866

### Decision letter and Author response
Decision letter https://doi.org/10.7554/eLife.56879.sa1
Author response https://doi.org/10.7554/eLife.56879.sa2

## Additional files

### Supplementary files
- Source data 1. TMA Source Data.
- Supplementary file 1. RAPID and Citrus Comparison.
- Supplementary file 2. Cell Subset Abundance and Population Totals per Patient.
- Supplementary file 3. Patient Characteristics.
- Supplementary file 4. CyTOF Panel.
- Supplementary file 5. Tumor Cell Abundance per Cell Subset.
- Supplementary file 6. Individual per-patient view of marker expression and subset abundance.
- Transparent reporting form

### Data availability
Annotated flow data files are available at the following link: https://flowrepository.org/id/FR-FCM-Z24K. Patient specific views of population abundance and channel mass signals for all analyzed patients in this study are currently available in Supplementary File 6. RAPID code is currently available on Github, together with example analysis data: https://github.com/cytolab/RAPID (copy archived at https://github.com/elifesciences-publications/RAPID).

The following dataset was generated:

| Author(s) | Year | Dataset title | Dataset URL | Database and Identifier |
|---|---|---|---|---|
| Leelatian N, Sinnaeve J, Mistry A, Barone S, Brockman A, Diggins K, Greenplate A, Weaver K, Thompson R, Chambless L, Moble B, Ihrie R, Irish J | 2019 | Unsupervised machine learning reveals risk stratifying glioblastoma tumor cells | https://flowrepository.org/id/RvFrKN2ctDJmmVNE4Z-nMJrAZeVraXbwvrhj-x3YaBZIV6n-WIanMrbhrVBx7yvODtX | FlowRepository, FR-FCM-Z24K |

The following previously published dataset was used:

| Author(s) | Year | Dataset title | Dataset URL | Database and Identifier |
|---|---|---|---|---|
| Good Z, Sarno J, Jager A, Samusik N, Aghaeepour, Simonds EF, White L, Lacayo NJ, Fantl WJ, Fazio G, Gaipa G, Biondi A, Tibshirani R, Bendall SC, Nolan GP, Davis KL | 2018 | Single-cell developmental classification of B cell precursor acute lymphoblastic leukemia at diagnosis reveals predictors of relapse | https://github.com/kara-davis-lab/DDPR/releases | Github Mass cytometry data for DDPR project, DDPR |

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
