## [Decision Letter]

**Acceptance summary:**

With larger datasets of single-cell data being made available, there is need in the field for analysis pipelines that can couple these experiments to clinical outcome and other variables in an unsupervised manner, and that can provide functional hypotheses that can be later tested in other cohorts in a simplified manner. Here, the authors have built RAPID, an algorithm that can take single-cell cytometry data as input and have extensively tested it in two cancer datasets, identifying populations of risk-stratifying cells. We envision that in years to come, RAPID will be able to identify markers correlated with important clinical characteristics, and that the functional relationships it finds and are confirmed will be useful for clinical practice.

**Decision letter after peer review:**

[Editors’ note: the authors submitted for reconsideration following the decision after peer review. What follows is the decision letter after the first round of review.]

Thank you for submitting your work entitled "High risk glioblastoma cells revealed by machine learning and single cell signaling profiles" for consideration by *eLife*. Your article has been reviewed by three peer reviewers, one of whom is a member of our Board of Reviewing Editors, and the evaluation has been overseen by a Senior Editor.

Our decision has been reached after consultation between the reviewers. Based on these discussions and the individual reviews below, we regret to inform you that your work will not be considered further for publication in *eLife*.

The reviewers, whose assessments are included below, agreed that RAPID is potentially interesting and useful to the scientific community but the lack of access to the code to test it prevents further assessment. In discussion, they agreed it is necessary for code and data to be accessible before publication for reviewers' perusal. Additionally, they considered the claim that biological findings "could be immediately used to guide clinical design" misguided, as an extensive comparison with existing literature has not been undertaken and extrapolates from results in a very small number of patients. Another suggestion for improving the work include analysing a larger validation cohort to establish the performance of RAPID in different datasets and under different conditions.

Reviewer #1:

In this work, Leelatian and collaborators study a set of 28 resected glioblastoma tissues from as many patients through single-cell technologies assessing 34 phospho-proteins, transcription factors and lineage proteins. They develop 2 technologies: 1) a set of 34 antibodies for single cell mass cytometry and 2) an unsupervised machine learning algorithm called RAPID – which is able to identify phenotypically similar cells ("clusters") and their association to survival variables.

I think the scale of the work is impressive (the study analyses >2 million cells) and the methodology seems very useful. However, there are a few issues that I think would need to be addressed before this manuscript is published.

1) I believe the software to be the main contribution of this manuscript, as the two proteins discovered at the end of the single-cell RAPID analysis have been studied in the context of glioblastoma/glioma and survival before (more of this below, but e.g., PMIDs 9445288, 28693199, 28885661, 27401156, 23719262). The data needs to be made publicly available, the same as the software. It says in the manuscript that it will be done so upon publication, but it absolutely needs to be released by the time the manuscript is published. At the moment it is impossible for me to assess whether RAPID may be easy to use, what variables it needs as input, and how easy it may be to install. Also, I do not know if the authors plan to release a detailed user manual, something that would be essential for publishing a piece of software.

2) I believe that the validation of the software needs to be done on more than one existing single-cell dataset, if possible. The fact that running different iterations of RAPID resulted in highly variable results in the same dataset (18 to 48 clusters identified), and that only one cluster overlapped in the OS vs PFS analyses calls, in my opinion, for more extensive testing. It's interesting that only 7 out of the 43 clusters identified when all cells were put together were considered "universal". Does this mean that this technique is highly susceptible to the number of tumours tested? Hopefully the software is easy to run and answers to these questions can be achieved relatively easily.

3) How do the authors reconcile their results with the observations by other studies that EGFR overexpression is associated with poor prognosis glioblastoma, seemingly contrary to the results in this study? (e.g. PMIDs 29445288, 28693199, 28885661). Linked to this point, I believe that the results are overhyped at times. For example, the phrase "These findings could be used immediately to guide clinical trial design" should be either removed or rewritten in a more measured way. Which population did the authors study, only European-descent (i.e. white) patients? Also, the number of samples is quite small for such a claim. Generalising in this way would be hurtful to global clinical practice.

4) About the classification of tumours into high/low for distinct markers. Two tumours may be highly similar and classified in different groups (in the example given in the text, tumours with 2.68% of cells in the cluster were classified as “low” but those with 2.7% as “high”). Would the conclusions be maintained if the high group was defined as those above the 75th percentile and the low group below the 25th percentile? How important is this definition to the results of the RAPID workflow?

Reviewer #2:

This is a potentially interesting manuscript that seeks to profile glioblastoma samples by mass cytometry to assess intra-tumoral heterogeneity at the protein level. The authors profile a large number of cells in 28 samples using 34 markers (retaining 26 markers in final set). They then use this information (with various filtering steps) to identify modules of variability within tumors and then use some of those modules/signatures to stratify patients’ outcome with a machine learning approach. They suggest that their findings can be "immediately" used to inform clinical trial design.

The study suffers from intrinsic limitations that are hard to overcome and that would be acceptable if the authors had not jumped to conclusions, they do not have the power to support. First, the analysis relies on 26 measured proteins, which is a rather limited set of parameters; any analysis that relies on protein measurement is also only as good as the antibody used and one should keep in mind that some of the clusters inferred might be technical; hence any major statement would require some form of validation by other approaches which is mostly lacking in this manuscript. But all that would be addressable/ok, if the authors had been more reasonable in their conclusions.

By far the main limitation and major caveat this reviewer sees is that the authors draw major conclusions on stratifying GBM patients for their outcome based on a cohort of 28 patients. This is massively underpowered to address patient stratification. The markers they home on are nothing very new (EGFR, SOX2, S100 etc) that have been interrogated in much larger cohorts (TCGA has >400 samples) and have not yielded as valuable information as claimed by the authors.

This study would only be useful if the 28 samples were a discovery cohort and their findings were validated in hundreds of patients’ samples. If EGFR was that good at predicting outcome of GBM, we would know by now. So unfortunately, underpowered study for the claims on outcome. The mass cytometry profile/data would be of interest if followed-up, but not for survival analysis given limited dataset.

Reviewer #3:

Leelatian, Sinnaeve et al. describes a single-cell mass cytometry data-set measuring 34 proteins and phospho-proteins across 28 glioblastoma patients. This represents a rich data-set which the authors take advantage to develop a novel computational tool, RAPID, to identify cell type clusters that inform on patient survival. This approach identified 9 cell clusters that stratified patients according to their overall survival, and these can be classified into negative and positive prognostic using S100B and EGFR protein abundance.

Although further work would be required to confirm the robustness and utility of these findings to the clinic, this study presents novel insights, tools and data-sets that will be of interest to the scientific community.

1) The separation and classification of the 43 cell clusters (Figure 1B) is not very clear. Consistently, these varied extensively across the 10 down-sample analyses with 18-48 optimal clusters. Have the authors tried multiple configurations of the tSNE parameters (e.g. perplexity, learning rate)? Clustering could also be compared to current state-of-the-art pipelines (PMID: 31217225), for example running Principal Component Analysis (PCA) as a pre-processing step before the non-linear tSNE.

2) Several clusters of glioblastoma cell types exist (Figure 2A), in particular a subset (lower left) seems to have a heterogeneous expression of Nestin and strong phosphorylation of AKT. Is this a common observation across other patient samples? If so, what is the percentage of cells this represents and could there be an impact on the analysis if this population, like the other cell types, were removed? Additionally, p-AKT is enriched in GNP cluster 37 and GPP cluster 41. Specifically, GPP cluster 41 is the only cluster enriched in 2 patients (LC03 and LC09) which are classified as GNP. To me this is counterintuitive, and might indicate some unique variation, either true biological or artefact, of this cell type.

3) The potential translation into the clinic seems to be one of the main messages of the manuscript and is indeed particularly interesting. Specifically, the fact that only S100B and EGFR expression is enough to stratify patient overall survival. Could the observations gained from the single-cell experiments be used to re-analyse bulk patient genomic and proteomic data-sets to support further the clinical relevance of these findings? For example, what is the percentage of glioblastoma *IDH1* wild type patients with S100B and EGFR high or low?

4) The availability of RAPID as a tool to the community is important and a positive aspect of this work. Thus, I believe the source should have been made accessible prior to publication.

[Editors’ note: further revisions were suggested prior to acceptance, as described below.]

Thank you for submitting your article "Unsupervised machine learning reveals risk stratifying glioblastoma tumor cells" for consideration by *eLife*. Your article has been reviewed by three peer reviewers, one of whom is a member of our Board of Reviewing Editors, and the evaluation has been overseen by Philip Cole as the Senior Editor. The following individual involved in review of your submission has agreed to reveal their identity: Julie Laffy (Reviewer #4).

The reviewers have discussed the reviews with one another and the Reviewing Editor has drafted this decision to help you prepare a revised submission.

We would like to draw your attention to changes in our revision policy that we have made in response to COVID-19 (https://elifesciences.org/articles/57162). Specifically, we are asking editors to accept without delay manuscripts, like yours, that they judge can stand as *eLife* papers without additional data, even if they feel that they would make the manuscript stronger. Thus, the revisions requested below only address clarity and presentation.

Summary:

In this revised manuscript, Leelatian, Sinnaeve and collaborators introduce the RAPID algorithm for identifying clusters of cells relevant for stratification of patient survival in single-cell datasets, especially mass cytometry datasets including more than 25 patients. In this new version, they have tested RAPID with two datasets from different cancer types, one generated by them and another one already published, and have performed extensive statistical validation both technical and biological.

Revisions:

Please address the following points raised by the reviewers:

1) Clustering is performed across all tumours combined, so the authors should distinguish inter-tumour effects from the intra-tumour effects that they describe. Could they show for example that the cluster phenotypes are derivable from a significant number of individual tumours? Inter-tumour effects would imply some different conclusions to the ones the authors draw, depending on the patient specificities of these effects. Assuming no patient specificity, one potentially interesting and even complementary conclusion would be that basal levels of certain proteins in a tumour are indicative of patient outcome (indiscriminately of any particular cellular subpopulation). Do the authors see that EGFR, S100B and other enriched proteins vary primarily within, or between, tumours? To check for patient specificity, the authors should state in the text i) which, if any, clusters were largely tumour-specific and ii) how many tumours contributed to each of the clusters. Without these figures, the biological relevance of each of the clusters (namely the NP and PP subsets) is difficult to assess. It might also be useful to add tSNE plots coloured by patient alongside the existing plots (Figures 1B, 2A, 4A). If there are indeed clusters dominated by individual tumours, I would think that they should be removed early on in the RAPID pipeline. If there are many such cases, then perhaps the authors would have to consider an alternative clustering approach.

2) Do any of the cell subsets identified (Figure 1B, C) reflect a proneural population? This is a known component of glioblastomas and has moreover been postulated to be associated with better prognosis because it is the dominant component in lower-grade gliomas (e.g. Verhaak et al., 2010). How do the authors reconcile this with their own findings? Or, if a proneural phenotype is lacking, could the authors comment on why that might be? One reason could be technical: simply that proneural markers were not sufficiently represented in the panel. If that is the case, the authors should acknowledge this in the text and take care not to overstate their results pertaining to the phenotypes of NP and PP subsets in GBM.

---

## [Author Response]

[Editors’ note: the authors resubmitted a revised version of the paper for consideration. What follows is the authors’ response to the first round of review.]

Reviewer #1:[…]1) I believe the software to be the main contribution of this manuscript, as the two proteins discovered at the end of the single-cell RAPID analysis have been studied in the context of glioblastoma/glioma and survival before (more of this below, but e.g., PMIDs 9445288, 28693199, 28885661, 27401156, 23719262).

We appreciate the reviewer’s enthusiasm regarding development of the software, and its ability to be used in the context of many different human diseases and we have updated the Discussion to include the references and to compare our results, especially for EGFR. Two key considerations for comparing our results to prior literature are: 1) Our analysis quantifies per-cell level of EGFR protein, as opposed to DNA mutation or RNA transcript in prior studies, and 2) we consider both EGFR and S100B together when scoring patients. Notably, the protein differences in EGFR and S100B revealed by mass cytometry were validated by IHC in a larger cohort.

The data needs to be made publicly available, the same as the software. It says in the manuscript that it will be done so upon publication, but it absolutely needs to be released by the time the manuscript is published. At the moment it is impossible for me to assess whether RAPID may be easy to use, what variables it needs as input, and how easy it may be to install.

All data and code were available to reviewers during the prior review, but we appreciate that the links were not apparent in the Supplement. We have updated the manuscript in several places to clarify how to access data. The link to Dataset 1 is:

https://flowrepository.org/id/RvFrKN2ctDJmmVNE4ZnMJrAZeVraXbwvrhjx3YaBZIV6nWIanMrbhrVBx7yvODtX

FlowRepository is public and enables deeper annotation. Dataset 2 and RAPID code are on Github and public here: https://github.com/cytolab/RAPID/. This includes RAPID source code and cytometry files from the published pre-B ALL example (Dataset 2).

Also, I do not know if the authors plan to release a detailed user manual, something that would be essential for publishing a piece of software.

The RAPID code and a detailed walkthrough with comments is provided as an R code and an R markdown file (at https://github.com/cytolab/RAPID) which includes instructions and guidelines within the code itself.

2) I believe that the validation of the software needs to be done on more than one existing single-cell dataset, if possible.

RAPID was used on an additional, single cell data set from B cell precursor acute lymphoblastic leukemia (Good et al., 2018, Dataset 2). The results from this test were last in the original Supplemental Figure 6 and have now been moved into the main text (Figure 2) and emphasized in the Results to highlight this important validation. We sought additional test datasets, but there were not published, annotated, single cell datasets that met the criteria for this study. We have noted this in the Discussion.

The fact that running different iterations of RAPID resulted in highly variable results in the same dataset (18 to 48 clusters identified), and that only one cluster overlapped in the OS vs PFS analyses calls, in my opinion, for more extensive testing. It's interesting that only 7 out of the 43 clusters identified when all cells were put together were considered "universal". Does this mean that this technique is highly susceptible to the number of tumours tested? Hopefully the software is easy to run and answers to these questions can be achieved relatively easily.

We appreciate the concerns regarding cluster variation and have added significant statistical testing to directly address this (see Statistical validation 1 and 2, especially). Note that we believe it is an advantage of RAPID to aim to be independent of user input and bias. Thus, we prefer to avoid having the user specify a target number of clusters, although it is possible to use the RAPID code in a way where the user can specify this number (and set a seed to achieve deterministic, invariable results). The additional statistical testing includes iterative FlowSOM runs, multiple runs of t-SNE, and comparison of subset features across many such runs to identify stable clusters and phenotypes. These results are now included as main Figure 3 and are graphically summarized in new panels in Figure 1C. Critically, while different absolute numbers of clusters were identified, the cellular phenotypes identified by RAPID from these clusters were consistent across runs and cell subsampling (Figure 3).

3) How do the authors reconcile their results with the observations by other studies that EGFR overexpression is associated with poor prognosis glioblastoma, seemingly contrary to the results in this study? (e.g. PMIDs 29445288, 28693199, 28885661).

We thank the reviewer for raising this concern and have updated the Discussion to compare and contrast our methods and findings to previous literature and have included these 3 EGFR citations.

Linked to this point, I believe that the results are overhyped at times. For example, the phrase "These findings could be used immediately to guide clinical trial design" should be either removed or rewritten in a more measured way. Which population did the authors study, only European-descent (i.e. white) patients? Also, the number of samples is quite small for such a claim. Generalising in this way would be hurtful to global clinical practice.

The reviewer makes a reasonable point, and we have tempered our language in this area, refocusing the manuscript on the use of RAPID as a discovery tool and referring to use in clinical research (as opposed to clinical practice). Regarding patient demographics, we confirmed that the overall survival characteristics were typical for the course of this disease in the United States patient population. Patient samples are reflective of the general demographics of patients at our institution (27 Caucasian patients and 1 African American patient).

4) About the classification of tumours into high/low for distinct markers. Two tumours may be highly similar and classified in different groups (in the example given in the text, tumours with 2.68% of cells in the cluster were classified as “low” but those with 2.7% as “high”). Would the conclusions be maintained if the high group was defined as those above the 75th percentile and the low group below the 25th percentile? How important is this definition to the results of the RAPID workflow?

We appreciate the point about the cutoff potentially being arbitrary and have address this directly in two ways. 1) In the revised manuscript we tried several cutpoints, including the 25%/75% approach suggested above, and found that the f-measure for patients falling into the same group is 0.86. 2) We also complemented the cutpoint version of the analysis with a continuous analysis using a multivariate Cox proportional-hazards model analysis that assessed GNP and GPP content as continuous features instead of using a cutpoint (now reported in the results).

Reviewer #2:This is a potentially interesting manuscript that seeks to profile glioblastoma samples by mass cytometry to assess intra-tumoral heterogeneity at the protein level. The authors profile a large number of cells in 28 samples using 34 markers (retaining 26 markers in final set). They then use this information (with various filtering steps) to identify modules of variability within tumors and then use some of those modules/signatures to stratify patients’ outcome with a machine learning approach. They suggest that their findings can be "immediately" used to inform clinical trial design.The study suffers from intrinsic limitations that are hard to overcome and that would be acceptable if the authors had not jumped to conclusions, they do not have the power to support.

We appreciate the concern that our wording was too strong for the data that we present. We have tempered our language throughout the manuscript to avoid over-stating conclusions and to focus on the utility of RAPID as a discovery tool that can generate hypotheses for further clinical research with other techniques (as in the IHC validation example included for Dataset 1).

First, the analysis relies on 26 measured proteins, which is a rather limited set of parameters; any analysis that relies on protein measurement is also only as good as the antibody used and one should keep in mind that some of the clusters inferred might be technical; hence any major statement would require some form of validation by other approaches which is mostly lacking in this manuscript. But all that would be addressable/ok, if the authors had been more reasonable in their conclusions.

The reviewer correctly points out that the validity of mass cytometry data, like other antibody-based approaches, is dependent on the quality of antibodies used for detection of features of interest. Our antibodies were rigorously validated and titrated on known controls prior to use on patient samples, and many of these antibodies have been published previously (Leelatian et al., 2017 and the companion protocol, Leelatian et al., 2017). We used IHC with different, validated antibody clones used routinely in our tissue pathology core to confirm the major findings (S100B v EGFR) in additional patient samples.

By far the main limitation and major caveat this reviewer sees is that the authors draw major conclusions on stratifying GBM patients for their outcome based on a cohort of 28 patients. This is massively underpowered to address patient stratification.

We appreciate the reviewer’s concern; for this reason, we included a tissue microarray with 73 cases for validation of the key biological features identified in the 28 dissociated samples used for mass cytometry analysis (Figure 5). In this case, the effect size of GNP/GPP prevalence is large relative to other studies reporting outcome stratification, and we were able to validate this stratification in an independent cohort.

The markers they home on are nothing very new (EGFR, SOX2, S100 etc) that have been interrogated in much larger cohorts (TCGA has >400 samples) and have not yielded as valuable information as claimed by the authors.

The reviewer correctly notes that EGFR, SOX2, and S100B have been investigated in GBM studies previously. The surprising finding in our work is not the specific features, all of which were included in our panel based on prior results, but that the abnormal co-expression of them at the protein level reveals new, previously unappreciated subsets, which also have distinct signaling phenotypes. The vast majority of GBM studies are based on measuring DNA and RNA, especially in bulk tumor samples, cell lines, or ex vivo conditions. We expect that transcript levels may give different results from protein measurements, especially when considered coordinately with other features rather than as a single variable. By combining primary patient samples from resected tumors and protein measurements, we sought to add to the field. We would note that the unexpected finding regarding EGFR revealing a subset of cells correlated with better outcomes, which was validated using traditional IHC, indicates how we can find unexpected and useful results with well-studied markers when assessing them at the single cell level using unsupervised analyses.

This study would only be useful if the 28 samples were a discovery cohort and their findings were validated in hundreds of patients’ samples. If EGFR was that good at predicting outcome of GBM, we would know by now. So unfortunately, underpowered study for the claims on outcome. The mass cytometry profile/data would be of interest if followed-up, but not for survival analysis given limited dataset.

We appreciate the reviewer’s concern regarding a pilot cohort size of N=28. For this reason, we tested whether the features most enriched on stable, risk-stratifying clusters found in our mass cytometry dataset could then stratify outcome in a separate, immunohistochemical analysis of N=73 glioblastoma cases. We agree that testing of the mass cytometry approach in a very large cohort would also be an exciting cancer biology advance, but it is beyond the scope of the present study focused on the RAPID algorithm.

Reviewer #3:[…]1) The separation and classification of the 43 cell clusters (Figure 1B) is not very clear.

We appreciate this feedback and have expanded both graphical depictions of our analysis (in Figure 1) and explanations in the text. In brief, RAPID employs a clustering algorithm called FlowSOM that builds self-organizing maps, based on t-SNE values, then groups nodes together based on similarity and the number of clusters input by the user. To avoid user bias in determining a cluster number, RAPID iteratively performed FlowSOM analyses from 5 to 50 clusters and chose a number that minimized intra-cluster variance for each feature. In Figure 1B, the clusters are projected back onto the t-SNE plot. We have also amended the Materials and methods to clarify this point.

Consistently, these varied extensively across the 10 down-sample analyses with 18-48 optimal clusters. Have the authors tried multiple configurations of the tSNE parameters (e.g. perplexity, learning rate)?

This question was addressed above in the reply to reviewer 1. Briefly, we have included a section on cell subsampling to directly address this concern.

Work in our own labs, as well as that by others (PMID: 31780669) suggest that increasing perplexity in t-SNE beyond optimized values, like those used here, does not significantly alter the results. We have tried different t-SNE parameters, run RAPID without using t-SNE, and run RAPID with UMAP (Figure 4), and the main results of the study are consistently observed.

Clustering could also be compared to current state-of-the-art pipelines (PMID: 31217225), for example running Principal Component Analysis (PCA) as a pre-processing step before the non-linear tSNE.

We appreciate the reviewer’s suggestion. PCA is routinely used as a pre-processing step in RNA-seq analysis prior to t-SNE implementation, due to the large number of genes measured, lack of dynamic range relative to noise, and preponderance of zero values (https://scikit-learn.org/stable/modules/generated/sklearn.manifold.TSNE.html). As a result of using targeted measurements and ionizing mass spectrometry, mass cytometry data do not have these issues (multiple log dynamic range, no zero / drop out, and a smaller number of highly impactful features measure). PCA analysis does not significantly reduce the dimensionality of mass cytometry data, as most of the measured features are non-redundant. Thus, PCA is not needed prior to t-SNE and, when used, does not significantly alter the resulting embedding. Notably, UMAP, which focuses more on global structure (more like PCA), returned similar results to the locally-oriented t-SNE analysis. Our workflow is otherwise comparable to the cited workflow, beginning with data clean up (which differs for sequencing data and cytometry data) followed by dimensionality reduction and clustering, (original modular workflow PMID: 25979346, reviewed in PMID: 27320317, updated regularly).

2) Several clusters of glioblastoma cell types exist (Figure 2A), in particular a subset (lower left) seems to have a heterogeneous expression of Nestin and strong phosphorylation of AKT. Is this a common observation across other patient samples? If so, what is the percentage of cells this represents and could there be an impact on the analysis if this population, like the other cell types, were removed?

The reviewer’s observation that several clusters of GBM cells exist is excellent and part of what inspired the development of RAPID to interrogate these clusters. The RAPID analysis will be impacted by which cells are included or excluded. To address this concern, we ran ten separate t-SNE analyses, subsampling cells each time (Statistical validation 2, Figure 3). In this revised manuscript, we have also included additional testing to assess how stable particular clusters and phenotypes are across multiple runs of t-SNE and FlowSOM. These improvements are highlighted in Figure 3.

In the specific case of Nestin/p-AKT, additional tumors in the cohort also have cells with this phenotype, though none as abundantly as LC26. The population is 7.61% of the LC26 tumor cells (9,220 cells out of 30,091). We also included patient-specific t-SNE maps with heat for all antigens, and % abundance for all clusters, as supplemental data (currently available as a PDF at https://www.biorxiv.org/content/10.1101/632208v3.supplementary-material).

Additionally, p-AKT is enriched in GNP cluster 37 and GPP cluster 41. Specifically, GPP cluster 41 is the only cluster enriched in 2 patients (LC03 and LC09) which are classified as GNP. To me this is counterintuitive, and might indicate some unique variation, either true biological or artefact, of this cell type.

The reviewer is correct in pointing out that p-AKT is enriched in GNP cluster 37 and GPP cluster 41. In our revised manuscript, cluster 41, while highly distinct when present, was not stable across multiple iterations of cell subsampling and therefore was removed from analyses of GNP/GPP. Relative to the other patients, LC03 and LC09 are enriched for cluster 41, but relative to their own tumor cell content, they have more GNP cells. We report that GNP cell content is the primary predictor of overall survival, suggesting that it is the balance of GNP and GPP cells that is indicative of a patient’s prognosis.

3) The potential translation into the clinic seems to be one of the main messages of the manuscript and is indeed particularly interesting. Specifically, the fact that only S100B and EGFR expression is enough to stratify patient overall survival. Could the observations gained from the single-cell experiments be used to re-analyse bulk patient genomic and proteomic data-sets to support further the clinical relevance of these findings? For example, what is the percentage of glioblastoma IDH1 wild type patients with S100B and EGFR high or low?

We appreciate the reviewer’s enthusiasm for applying our findings and envision the results being widely applied to current or future GBM data sets and in clinical research. However, bulk analysis techniques which aggregate signals from all cells may lack resolution needed to observe key findings, especially in tumors with large proportions of infiltrating immune cells. Currently, there are many fewer samples in TCGA (N=50) for which bulk proteomic data and outcome are available – notably, fewer than the number included in the validation set here (N=73).

4) The availability of RAPID as a tool to the community is important and a positive aspect of this work. Thus, I believe the source should have been made accessible prior to publication.

All data and code were available to reviewers during the prior review at https://flowrepository.org/id/RvFrKN2ctDJmmVNE4ZnMJrAZeVraXbwvrhjx3YaBZIV6nWIanMrbhrVBx7yvODtX. We have updated the manuscript in several places to clarify how to access data. RAPID code and cytometry files from the BCP-ALL example are publicly available on Github. The other FCS files from the glioblastoma study are available on FlowRepository using the above link.

[Editors’ note: what follows is the authors’ response to the second round of review.]

Revisions:Please address the following points raised by the reviewers:1) Clustering is performed across all tumours combined, so the authors should distinguish inter-tumour effects from the intra-tumour effects that they describe. Could they show for example that the cluster phenotypes are derivable from a significant number of individual tumours?

We appreciate this point and have highlighted two key results: 1) cell clusters were observed in multiple tumors and 2) multiple tumors contributed to cell clusters (the reviewer’s question). Thus, there were not specificity relationships between clusters and patients. In this analysis, there were not “private” clusters containing cells from only one tumor.

First, Row 36 of Supplementary file 2 indicates the number of tumors contributing at least 1% of that tumor’s cells to a cluster. For example, 14 tumors had more than 1% of their events assigned to GPP clusters and 13 tumors had more than 1% of events assigned to GNP clusters.

Second, a new Supplementary file 5 has been added to address the reviewer’s question directly by showing the percent of each cluster that was derived from each tumor. Row 32 indicates the number of tumors contributing at least 1% of that cluster’s cells. Notably, the observed clusters were all derived from multiple tumors (at least 4 and a median of 12 tumors contributed to clusters).

Inter-tumour effects would imply some different conclusions to the ones the authors draw, depending on the patient specificities of these effects. Assuming no patient specificity, one potentially interesting and even complementary conclusion would be that basal levels of certain proteins in a tumour are indicative of patient outcome (indiscriminately of any particular cellular subpopulation). Do the authors see that EGFR, S100B and other enriched proteins vary primarily within, or between, tumours?

EGFR and S100B, and many of the features measured, vary both within individual tumors and between patients. Notably, the single cell study design allows the reader to see that these variations are driven by distinct cell types in which proteins were co-expressed (or simultaneously absent). The comparison of the single cell cytometry and imaging makes it clear that this tracking of cell subsets revealed by multiple markers is more sensitive than tracking the markers as separate features, but that the signal from the cells is strong enough that the traditional unimodal analysis can provide a rough surrogate once the key features are known. The variation from patient to patient can also be seen in Supplementary file 6, which shows cell event distribution and “heat” for each patient and parameter on a common (all 28 patients) t-SNE plot, and in our tissue microarray analysis.

To check for patient specificity, the authors should state in the text i) which, if any, clusters were largely tumour-specific and ii) how many tumours contributed to each of the clusters. Without these figures, the biological relevance of each of the clusters (namely the NP and PP subsets) is difficult to assess.

We appreciate this point and have added Supplementary file 5 and highlighted Supplementary file 2, which together indicate no specificity relationships. We have added the following sentences to the manuscript to highlight this point (found in the Results section titled: Identification of risk stratifying glioblastoma cells in Dataset 1):

“The number of tumors that contributed to each cluster varied between the 43 clusters, but a median of 8 tumors contained cells in each cluster (Supplemental file 2, Supplementary file 6). Furthermore, each cluster contained cells from at least 4 tumors and, at the median, contained cells from 12 tumors (Supplemental file 5, Supplementary file 6).”

It might also be useful to add tSNE plots coloured by patient alongside the existing plots (Figures 1B, 2A, 4A). If there are indeed clusters dominated by individual tumours, I would think that they should be removed early on in the RAPID pipeline. If there are many such cases, then perhaps the authors would have to consider an alternative clustering approach.

The 28-page Supplementary file 6 was included to help address the point the reviewer is raising. This supplement shows the requested data in a per-patient view (one page per patient). Along with Supplemental file 2 and the added Supplemental file 5, we believe the reviewer’s concern is well addressed: specificity relationships between clusters and patients were not observed and clusters were not “private” to a specific tumor. Notably, coloring cells in the combined t-SNE by patient results in a view where the 131,880 cell dots outnumber the pixels and obscure each other, making it difficult to see where a given patient falls on the t-SNE axes.

2) Do any of the cell subsets identified (Figure 1B, C) reflect a proneural population? This is a known component of glioblastomas and has moreover been postulated to be associated with better prognosis because it is the dominant component in lower-grade gliomas (e.g. Verhaak et al., 2010). How do the authors reconcile this with their own findings? Or, if a proneural phenotype is lacking, could the authors comment on why that might be? One reason could be technical: simply that proneural markers were not sufficiently represented in the panel. If that is the case, the authors should acknowledge this in the text and take care not to overstate their results pertaining to the phenotypes of NP and PP subsets in GBM.

The proneural, mesenchymal, and classical subtypes are largely defined by DNA alterations or transcript expression, while this paper is focused on per-cell measurements at the protein level. We did include measurements of proteins which might be expected to be enriched in each of these subclasses, including PDGFRA and SOX2 (expected to be found in proneural tumors, as described by Verhaak et al., 2010). We observed many cell subpopulations with SOX2 protein expression, including several GNP subsets, but did not observe co-enrichment of PDGFRA protein in these cells. Within the Discussion, we explore potential reasons why protein-level data may reveal different features than studies of DNA or expressed RNA transcripts. It is also important to note that the referenced classification has been updated; when IDH-mutant tumors are excluded, as in our study, proneural tumors are similar to other subclasses in outcome (Wang et al., Cancer Cell 2017).